# Milli-scale cellular robots that can reconfigure morphologies and behaviors simultaneously

Xiong Yang [1], Rong Tan[1], Haojian Lu [1,2], Toshio Fukuda[3] & Yajing Shen [1,4,5 ✉]

Modular robot that can reconfigure architectures and functions has advantages in unpredicted environment and task. However, the construction of modular robot at small-scale remains a challenge since the lack of reliable docking and detaching strategies. Here we report the concept of milli-scale cellular robot (mCEBOT) achieved by the heterogeneous assembly of two types of units (short and long units). Under the magnetic field, the proposed mCEBOT units can not only selectively assemble (e.g., end-by-end and side-by-side) into diverse morphologies corresponding to the unstructured environments, but also configure multi-modes motion behaviors (e.g., slipping, rolling, walking and climbing) based on the on-site task requirements. We demonstrate its adaptive mobility from narrow space to high barrier to wetting surface, and its potential applications in hanging target taking and environment exploration. The concept of mCEBOT offers new opportunities for robot design, and will broaden the field of modular robot in both miniaturization and functionalization.

[1] Department of Biomedical Engineering, City University of Hong Kong, Hong Kong, China. [2] State Key Laboratory of Industrial Control and Technology, and Institute of Cyber Systems and Control, Zhejiang University, Hangzhou 310027, China. [3] Department of Micro-Nano Systems Engineering, Nagoya University, Nagoya, Japan. [4] Shenzhen Research Institute of City University of Hong Kong, Shenzhen 518057, China. [5] Department of Electronic and Computer Engineering, The Hong Kong University of Science and Technology, Clear Water Bay, Kowloon, Hong Kong, China. ✉email: eeyajing@ust.hk

Untethered small-scale robots have been receiving increasing attention for micromanipulation, microfactory, and biomedical applications owing to their remote actuation and non-invasive accessing ability in the narrow-enclosed environment[1–4]. By far, scientists have proposed versatile small-scale robots with different architectures, and demonstrated their potentials in handling different tasks[5–8]. For instance, the micro robots with helical structure[9,10] are able to achieve 3D navigation in large-span viscosity liquid, the film-like milli robots fabricated by soft material[11,12] can perform multimodal motion in harsh environment, and the micro granular robots with chemical or physical modifications[13,14] are capable of target drug delivery in human body. Despite the excellent locomotion ability, these robots are usually designed as a whole unit and are difficult to transform, reconstruct, or scale-up to new morphologies and behaviors to conduct diverse tasks.

Distinct from the conventional robot construction concept, the modular robot is a collection or assembly of several autonomous robotic units, where these separated units can connect each other to act as a whole entity, and more importantly, can reconfigure its morphologies and behaviors to adapt to different environments and tasks. Since the concept of cellular robot (CEBOT) in 1988[15], as a long-term goal with exciting potential for payoffs, scientists have made lots of efforts in the modular robot, including compact unit design[16,17], reliable connection components[18,19], effective algorithms coordination[20–22], and on-site extensibility[23,24]. However, under the critical constraints of space and energy consumption, constructing modular robot at small-scale is still very difficult limited by the lack of conscious approaching, diverse docking, and selective detaching strategies between micro units (Table 1). Although swarm design can cluster massive micro/nano particles to perform certain collective behaviors[25–27], drawbacks in poor monomer control, non-selective unit approaching, and virtual modular connection make it only works under the global magnetic actuation. Moreover, despite the diverse achievable patterns of swarm, its motion behaviors are limited and mostly confined to the liquid environment. Applying additional frameworks, restriction or assistance[28–33] can also collect or assemble small units as desired, however, these strategies impose more or less restrictions on the reversibility of assembly and the application of robot. Furthermore, most of these strategies only focus on the morphological reconfiguration and ignore the diversity of behaviors and functions of robot, which makes the small-scale modular robot far from reliable in practice.

In this article, we present a milli-scale CEBOT (mCEBOT) construction strategy that can reconfigure morphologies and behaviors simultaneously (Fig. 1a). Since the unique design of mCEBOT units with soft magnetic material, frustum shape, and slightly different dimensions, the reversible, controllable and multiplex heterogeneous assembly is achieved. We demonstrate the adaptability of mCEBOT to unstructured environments with narrow space, high barrier, wetting surface, and hanging target, by reconfiguring to independent monomer, bastinade shape, biped structure, and hoe mode for slipping, rolling, walking, and climbing respectively. Moreover, benefiting from the heterogeneous assembly endowed selective detaching and recycling of units, our mCEBOT can stepwise unload some units for path marking during the exploration of complex environments. We believe that the proposed mCEBOT will broaden the field of modular robot in both miniaturization and functionalization, and will shed new light on the realization of flexible, adaptive, and functionalized robot to tackle complicated environments and tasks at small-scale.

## Results

**Units design and characterization.** To realize the conscious assembly, on-demand separation and partial unload of mCEBOT,

**Table 1 The comparation of existing assembly methods for small-scale modular robot.**

| | Independent control units | Reversibility of assembly | Morphologies reconfiguration | Behaviors reconfiguration | Features | References |
|---|---|---|---|---|---|---|
| Self-assembly | - | Poor | Poor | Poor | Assembly process is uncontrollable | 34,35 |
| Irreversible assembly | - | - | - | Poor | Non-reversing | 36,37 |
| Virtual assembly | - | Good | Good | Good | Global actuation and limited to liquid environment | 25–27 |
| Assisted assembly | Good | Poor | Good | Poor | Less diversity in behaviors and functions | 28–33 |
| Heterogeneous assembled mCEBOT | Good | Good | Good | Good | Reconfigure morphologies and behaviors simultaneously | This work |

"-" means cannot achieve; "Poor" means few can achieve; "Good" means most can achieve.

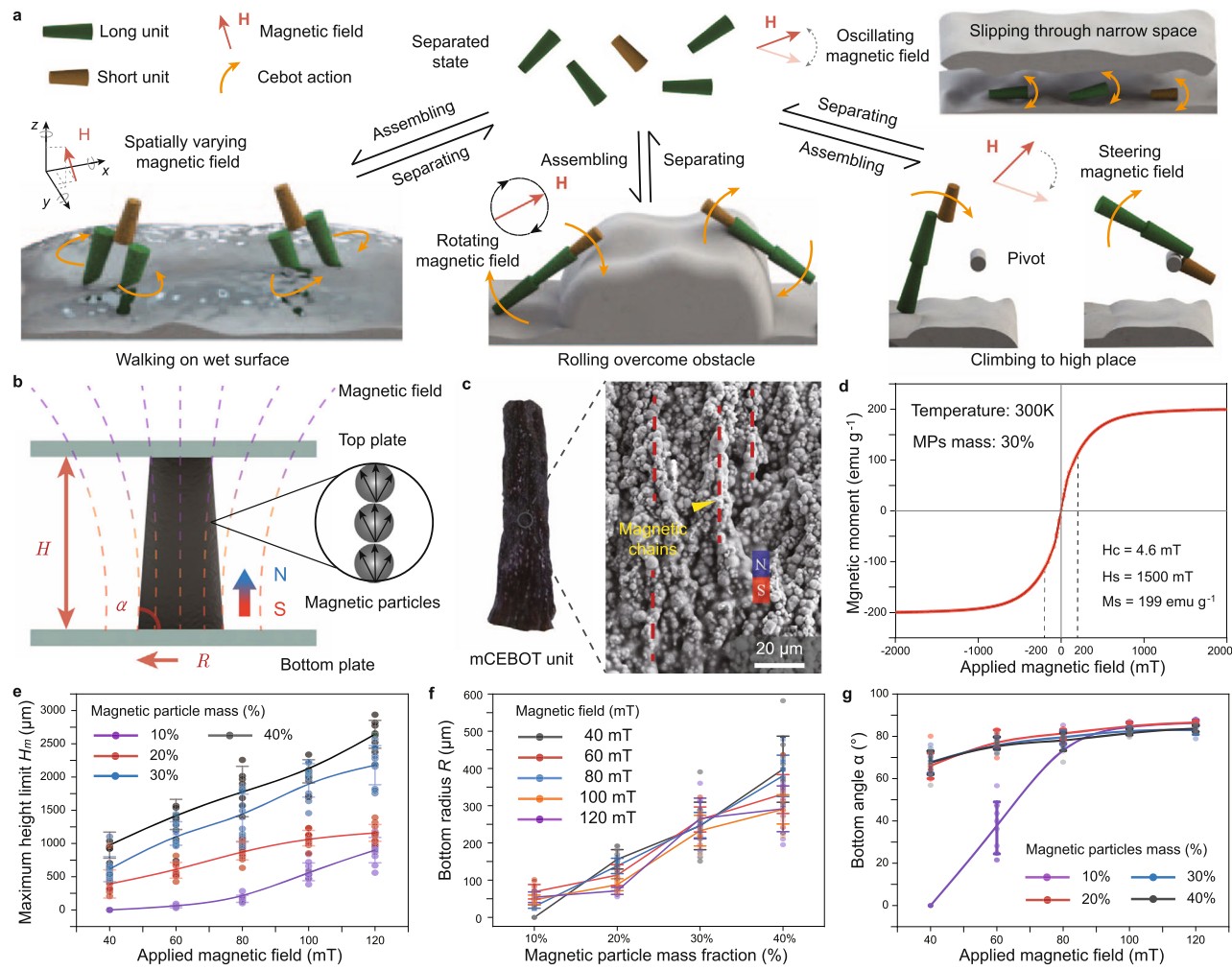

**Fig. 1 Design, fabrication and characterization of mCEBOT units. a** Schematic of mCEBOT which can reconfigure architectures and functions according to the unstructured environment and tasks. **b** Magnetic-field-assisted fabrication for mCEBOT units. **c** Surface texture of mCEBOT unit under SEM. **d** Magnetic susceptibility of mCEBOT unit under the magnetic field from −2000 to 2000 mT. **e** The relationship between unit's maximum height and applied magnetic field with different magnetic particle mass from 10 to 40%. **f** Bottom width of unit corresponding to various magnetic particle mass from 10 to 40% under magnetic field from 40 to 120 mT. **g** The relationship between unit's bottom angle and applied magnetic field with different magnetic particle mass from 10 to 40%. Error bars indicate the standard deviation for $n = 10$ measurements at each data point.

we take soft magnetic material and biocompatible Eudragit polymer for fabrication. Particularly, we design the units to frustum shape with a large aspect ratio and two differentiated section radiuses. Here the large aspect ratio of unit could guide the magnetic poles distributing to two ends to reduce the probability of unpredictable and uncontrollable assembly. And the two differentiated section radiuses can endow unequal arrangements gap during multiple side-by-side assemblies for reducing internal magnetic repulsion, which enables the easier and more stable connection than the uniform cylinder with the same size (Supplementary Fig. 1 and Supplementary Note 1). To obtain the raw material of mCEBOT, 30% Eudragit solution is firstly prepared by adding Eudragit polymer into pure alcohol, then mixing with iron particles at a mass ratio of 7:3. To mass-produce the frustum-shaped units, the modified magnetic-field-assist manufacturing is adopted (Fig. 1b and Supplementary Fig. 2a). Here, we transfer the raw material onto a non-magnetic plane and flatten it by spin coating, following cover a top non-magnetic plane with the desired distance. After that, a vertical magnetic field is applied and the frustum-shaped units will grow from the bottom substrate to top substrate under the combined action of magnetic field, surface tension, and gravity (Supplementary Fig. 2b).

Finally, the designed units are obtained by extracting them from the plate after curing.

Benefiting from the large aspect ratio of frustum shape and the magnetic-field-assist manufacturing, the obtained unit has a directional easy magnetization axis. As the SEM image shown in Fig. 1c, the iron particles inside the unit are well aligned along its long axis to form the magnetic chains coinciding with the direction of the applied magnetic field lines. To investigate the magnetization property quantitatively, we further measure the magnetic moment density of units under changing magnetic field from −2000 to 2000 mT (Fig. 1d). The results show that the unit has small residual magnetization (magnetic coercivity ~4.6 mT) when the magnetic field strength is down to 0 mT due to the soft magnetic property of the iron particles. Moreover, the magnetic moment is nearly linear (~0.56 emu $g^{-1}$ $mT^{-1}$) under the actuation region (−200 to 200 mT) in practice. Such directional, erasable and controllable magnetization makes the stable actuation, assembly and separation of mCEBOT under magnetic field possible (Supplementary Note 2 and Supplementary Note 3).

To obtain the uniform and replaceable units controllably, we investigate the key parameters during fabrication, mainly including height $H$, bottom radius $R$, and angle $\alpha$. Although the

actual height $H$ of unit is forcedly equal to the distance $D$ between the bottom and top plates, its achievable maximum growth height $H_m$ could be different under the different fabrication conditions. To further understand the fabrication process, we first measure the maximum growth height $H_m$ of unit with magnetic particle mass from 10 to 40% under the magnetic field with strength from 40 to 120 mT without top substrate limit. The results suggest $H_m$ is proportional to both the magnetic particle mass and applied magnetic field strength, which changes from nearly 0 μm (10% magnetic particle mass and 40 mT magnetic field) to ~2750 μm (40% magnetic particle mass and 120 mT magnetic field) (Fig. 1e). For the bottom radius $R$, regardless of the magnetic field strength, its average value and distribution range both increase as the magnetic particle mass changes from 10 to 40% (from 0 to ~100 μm in 10% magnetic particle mass, from ~200 to ~500 μm in 40% magnetic particle mass), indicating that the bottom radius $R$ is mainly determined by the magnetic particle mass but has no significant relationship with the applied magnetic field (Fig. 1f). On the contrary, curves in Fig. 1g indicate the bottom angle $\alpha$ increases slowly from ~68° to ~85° as the applied magnetic field increases from 40 to 120 mT, while expressing no significant relationship with the magnetic particle mass. Note that for the fabrication of unit with 10% magnetic particle mass, the bottom angle $\alpha$ is obviously abnormal due to the too weak magnetic field (less than 80 mT), which will not be adopted in practical manufacturing.

We finally choose 30% magnetic particles mass and 100 mT magnetic field for all units' fabrication after comprehensively considering the unit size and actuation ability. Under these conditions, the obtained units owe the same bottom radius (~200 μm), bottom angle (~83°), and magnetization (~55 emu g$^{-1}$ under 100 mT). While the heterogeneous dimensions between short and long units are achieved by adjusting the distance $D$ of substrates during fabrication, which are ~850 and ~1200 μm respectively in this manuscript.

**Controllable and multiplex heterogeneous assembly**. The on-demand assembly and separation of units are realized by the magnetization and demagnetization of soft magnetic material through applying and removing magnetic field respectively. As the simulation results shown in Fig. 2a, when the frustum-shaped units are placed in the magnetic field, all of them will arrange along the direction of the magnetic field and be magnetized with the magnetic poles at two ends. By adjusting the relative position and distance among the units, we can achieve two stable assembly states controllably (Fig. 2b), i.e., tending to end-by-end connection when the horizontal distance $x$ between two neighboring units in long axis direction is larger than 0 and tending to side-by-side connection while $x < 0$. Benefiting from the non-contact triggering and no specific connection mechanism, the magnetic attraction-based assembly between units can tolerate large position errors even under harsh surface condition. On the other hand, the assembled whole unit can be easily separated benefited from the soft magnetic property, since the magnetization and corresponding magnetic attraction force between units will reduce to almost 0 once the external magnetic field is removed. Further, to eliminate the potential electrostatic force, an oscillation magnetic field with decreasing strength (from 30 to 0 mT) could more effectively separate the connected units.

Due to the global actuation of magnetic field in the small area, it's very challenging to manipulate a single unit while does not affect neighboring units. During our heterogeneous assembly, the controllable adjustment of the relative position between short and long units is enabled, because their heterogeneous dimensions make them perform different step sizes and different trajectories even under the same magnetic field. As shown in Fig. 2c, the short unit with height $H_1$ and the long unit with height $H_2$ are located in initial point "$O$" and "$O'$" respectively. Theoretically, their next foothold under rolling motion must be located at a point on the circle (dotted line) with their initial points and lengths as the center and radius respectively. Since the motion direction of short and long units are always consistent under the same magnetic field, their movement difference $d$ in each step, ranging from $-|H_2-H_1|$ to $|H_2-H_1|$, can be calculated according to the units' geometric and motion direction. Thus, by path planning, the long unit can finally approach the short unit for assembling after several steps. It's worth noting that the newly assembled unit naturally has a different dimension from the original long unit, making the selective and controllable assembly can be implemented continuously. The detailed assembly strategy and trajectory planning for end-by-end and side-by-side connection can be found in Supplementary Fig. 3, Supplementary Note 4, and Supplementary Note 5.

To evaluate the stability of heterogeneous assembly, we investigate the bonding force between units with end-by-end and side-by-side connections respectively. As the results shown in Fig. 2d, both connection methods express the index increase in bonding force as the applied magnetic field increases, and the bonding force of end-by-end connection is approximately two times larger than that of side-by-side connection. In detail, the bonding force of end-by-end connection increases from ~4.0 μN under 50 mT to ~25.5 μN under 100 mT, while increases from ~1.5 μN under 50 mT to ~12.5 μN under 100 mT for the side-by-side connection. Note that a stronger magnetic field can be adopted in case more stable connection or more effective actuation is required. In this manuscript, the micro newton connection force under 100 mT magnetic field is adequate to ensure the stable movement of mCEBOT in different environments as demonstrated in the following sections.

**Multiple configurations of morphologies**. To intuitively reveal the effectiveness of heterogeneous assembly in mCEBOT's morphologies configurations, we employ one short and two long units to embody the bastinade shaped and biped robot sequentially (Supplementary Movie 1). As the time-lapse images shown in Fig. 3a, three units distribute in random and point in the same direction as the magnetic field at 0 s. Then, the 100 mT rotating magnetic field is applied to actuate the long unit to approach the short units. Under effective path planning, the first and second end-by-end connections are achieved at 22.5 and 39.5 s respectively, and three separated units are configured as the bastinade-shaped mCEBOT. To verify the effectiveness of the bastinade shape as a whole unit, we adjust the magnetic field direction to be perpendicular to the plane at 78.5 s. As the other end of the robot gradually leaves the ground and eventually becomes perpendicular to the plane, the assembly, and actuation of bastinade-shaped mCEBOT as a whole unit are proved to be feasible. At last, by decreasing the magnetic strength to 0 mT and following a low intensity oscillating magnetic field, the obtained bastinade shape separates into three individual units again at 100.0 s. Similarly, the short unit can also connect with long units side-by-side (at 4.0 and 80.5 s) to stand up as a biped mCEBOT (at 90.5 s) and separates on-demand (at 123.5 s) (Fig. 3b).

Note that the end-by-end and side-by-side connection strategies are combinable and expandable to achieve multiple configurations (Fig. 3c and Supplementary Fig. 4). As the demonstrations shown in Fig. 3d, by changing the combination and sequence of two types of connection methods, multiple configurations of mCEBOT can be achieved under the controllable action of magnetic field. Here the 5 units assemble by

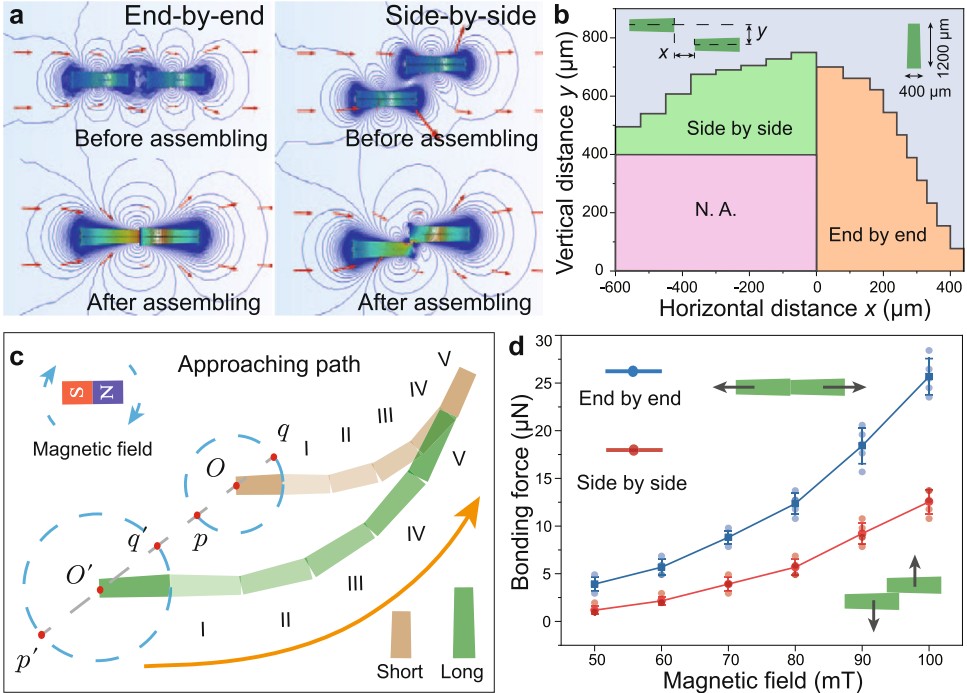

**Fig. 2 Heterogeneous assembly between mCEBOT units. a** Simulation results under magnetic field, where the neighboring units connect by end-by-end or side-by-side according to their relative position and distance. **b** Distribution of connection states between two identical long units under 100 mT magnetic field. **c** Heterogeneous assembly between short and long unit by taking advantage of their different step size. **d** Bonding force between two long units which connected by end-by-end and side-by-side respectively. Error bars indicate the standard deviation for $n = 5$ measurements at each data point.

side-by-side as the state 1 at initial, then separate into independent monomers at 5 s by removing external magnetic field for further architecture reconfiguration. Benefiting from the controllable and selective heterogeneous assembly, these individual units reassemble into the state 2 at 60 s by combining two connection methods. Similarly, repeating the controllable separation and selective assembly, the 5 units can reconfigure as state 3 at 120 s by end-by-end. Besides that, the controllable and selective heterogeneous assembly can also enable the construction of single mCEBOT (Fig. 3e) or multiple mCEBOTs (Fig. 3f) in a large number of units. When we continue to assemble the obtained assembly as a whole unit with other units, we can construct a single mCEBOT with a large number of units. While, when we re-select new units to start the assembly of a new mCEBOT after the previous mCEBOT has been assembled, we can construct multiple mCEBOTs. This controllable and selective heterogeneous assembly provides more options for the construction of mCEBOT.

**Motion behaviors and mechanisms of typical configurations**. The actuation of mCEBOT is achieved by applying an external magnetic field, during which the motion behaviors are highly relevant to its architecture. Despite the diverse achievable morphologies, they can be classified into four typical configurations, i.e., independent monomer (Fig. 4a), bastinade shape (Fig. 4b), biped structure (Fig. 4c), and hoe mode (Fig. 4d), with the attributes of the smallest indivisible unit, largest aspect ratio, multi-point contact, and hook anchor, respectively. In the following, we will elucidate the motion behaviors and the corresponding magnetic driving mechanisms of these four typical configurations.

The independent robot monomer is the smallest indivisible unit, therefore we configure the most fault-tolerant and anti-interference slipping as its motion behavior. As mechanical analysis shown in Fig. 4a, the independent robot monomer will align its long axis following the direction of magnetic field under

the action of magnetic torque. Under the oscillating magnetic field, the independent robot monomer will swing with the applied magnetic field direction and its two ends will contact the ground alternately. During which, the independent robot monomer will slip forward driven by an instantaneous acceleration when resistance force $f_x$ is smaller than the horizontal magnetic pulling force $F_{Mx}$. As the motion trajectory shown in Fig. 4e, under the oscillating magnetic field with a strength of 100 mT, a frequency of 1 Hz, and an oscillating angle from $-20°$ to $20°$, the independent robot monomer can slip forward about 28% of its body length in one gait cycle and express height fluctuation about 34% of its body length in the vertical direction. The details about the slipping motion of independent robot monomer can be found in Supplementary Fig. 5, Supplementary Movie 2, and Supplementary Note 6.

For the bastinade shaped mCEBOT with a large aspect ratio, we configure the rolling as its motion behavior by applying the continuously rotating magnetic field to guarantee the smooth and stable movement of the bastinade shape. As illustrated in Fig. 4b, when the magnetic field rotates, the robot will also rotate around the contact point, since its easy magnetization axis tends to follow the magnetic field direction. Based on that, the bastinade-shaped mCEBOT can roll forward by adopting its two ends as fulcrum alternately under the action of continuously rotating magnetic field. Different from the oscillating magnetic field for slipping, the rotating magnetic field for rolling is smooth, unidirectional and continuous, which can not only keep the assembled architecture stable but also prevent the robot from unexpected skidding. As the motion trajectory shown in Fig. 4f, under the rotating magnetic field with a strength of 100 mT and a frequency of 0.25 Hz, the rolling motion shows an attainable height fluctuation of 100% of itself body length and an effective step size of 200% of itself body length in one gait cycle. More details about rolling motion of bastinade-shaped mCEBOT can be found in Supplementary Fig. 5, Supplementary Movie 2, and Supplementary Note 7.

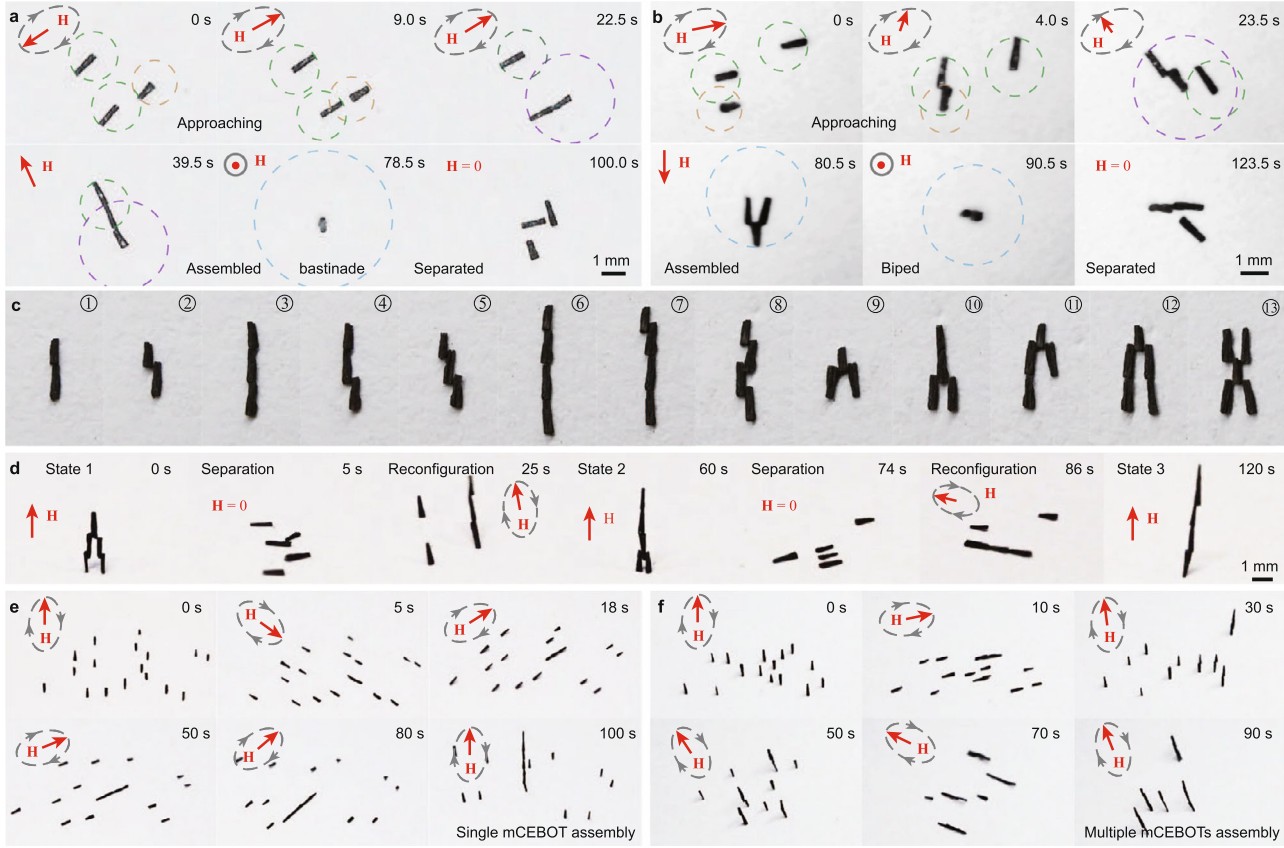

**Fig. 3 Multiple configurations of mCEBOT morphologies. a** Assembly of bastinade-shaped mCEBOT by end-by-end connection and corresponding separation by removing magnetic field. **b** Assembly of biped mCEBOT by side-by-side connection and corresponding separation by removing magnetic field. **c** Protean morphologies achieved by introducing units' number and combining two connection methods. **d** Multiple configurations of mCEBOT's morphologies by controllable assembly and separation. **e** The construction of single mCEBOT in a large number of robotic units. **f** The construction of multiple mCEBOTs at the same time in a large number of robotic units.

For the mCEBOT with biped structure, we configure the walking as its motion behavior considering its irregular shape and multi-point contact. As illustrated in Fig. 4c, to convert the leg spacing into effective step size, we adjust the spatial attitude of robot by magnetic field direction to make its legs contact with ground and work as a fulcrum for stepping forward alternately. During which, the magnetic torque $T_{My}$ along the y axis will tilt the body of biped structured mCEBOT and lift one foot with the other foot as the fulcrum. Under the combined action of the magnetic torques $T_{Mx}$ and $T_{Mz}$ along the x and z axis respectively, the robot body rotates around the fulcrum to convert the leg spacing into effective step size, and the continuous walking can be achieved by alternating two feet as the fulcrum. As the motion trajectory shown in Fig. 4g, under the spatially varying magnetic field with a strength of 100 mT, the biped structured mCEBOT can controllably step forward with a unilateral step size of about 75% of its leg spacing. The details about walking motion of biped structured mCEBOT can be found in Supplementary Fig. 5, Supplementary Movie 2, and Supplementary Note 8.

Under the hoe mode, we configure the climbing as motion behavior of mCEBOT by applying the steering magnetic field. As illustrated in Fig. 4d, the unaligned part of side-by-side assembly can work as hook for anchoring pivot. During which, the combined action of magnetic pulling $F_M$ and gravity $mg$ make the hook can always contact with the pivot while keep the pivoting ability. Based on that, the robot can climb with the pivot as a fulcrum under the magnetic torque $T_{My}$ when the magnetic field steers. By using the advantages of the external pivot, mCEBOT under the hoe mode can reach a high place that is even higher

than its own body length. As the motion trajectory shown in Fig. 4h, the maximum height of the climbing motion can reach 142% of itself body length when the magnetic field with a strength of 100 mT steering 160°. More details about climbing motion of hoe-shaped mCEBOT can be found in Supplementary Fig. 6, Supplementary Movie 2, and Supplementary Note 9.

**Locomotion adaptability in unstructured environment.** Benefiting from the reconfigurable morphologies and the corresponding motion behaviors, our mCEBOT can adapt to diverse environments, e.g., narrow channel, wet surface, high barrier and hanging target (Fig. 4i–l and Supplementary Note 10). For the movement in narrow channel, the dimension and locomotion of the robot are strictly restricted by the limited space. Compared with the conventional robots, mCEBOT can separate itself into tiny individual units and adopt the slipping pattern for locomotion. Benefited from the tiny size of unit and the small height fluctuation of slipping, the robot monomer can cross the narrow channel with a minimum diameter almost the same as its own diameter theoretically (Fig. 4i and Supplementary Movie 3). For the high barrier, it can be addressed by assembling numbers of units into bastinade shape and configuring rolling as motion behavior. In this case, the end-by-end connection can increase the height of mCEBOT to the maximum extent to overcome the obstacle through rolling. As shown in Supplementary Fig. 7a, initially, the independent robot monomers cannot overcome the barrier (height 2 mm) due to the height difference and the barrier's slippery surface. In contrast, after the individual units are

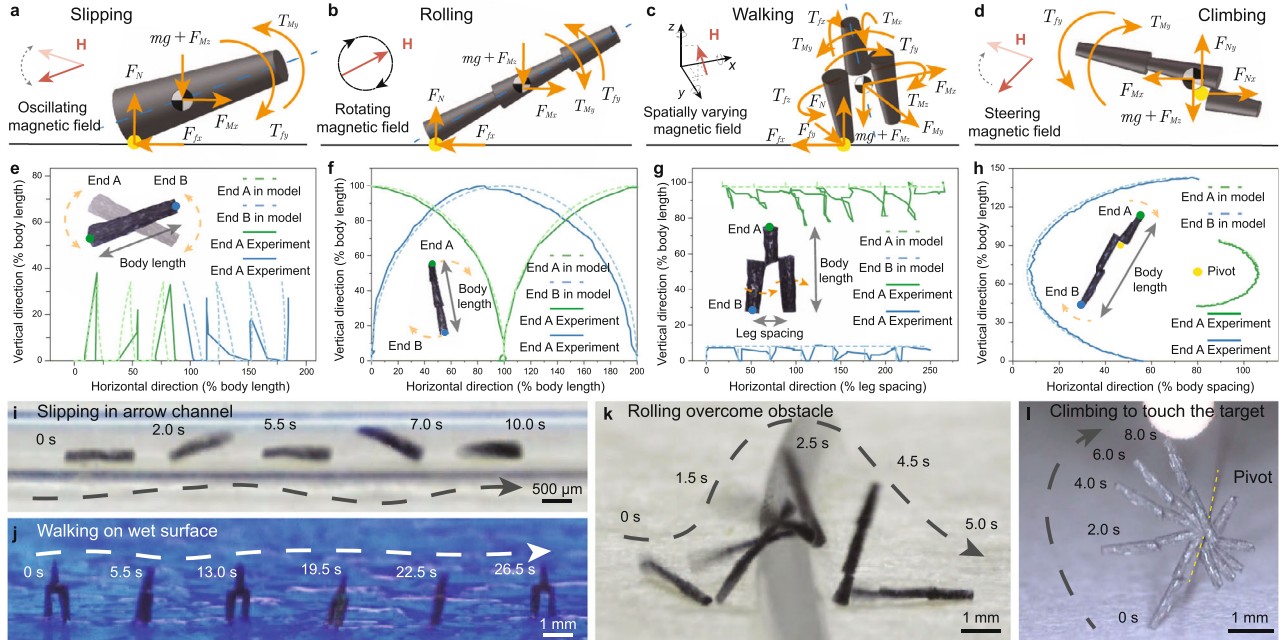

**Fig. 4 Diverse morphologies and corresponding motion behaviors of mCEBOT. a** Slipping of mCEBOT unit in the separated state under oscillating magnetic field. **b** Rolling of bastinade shaped mCEBOT as a whole unit under rotating magnetic field. **c** Walking of biped mCEBOT in the assembled state under the magnetic field with spatial variation of direction. **d** Climbing of hoe shaped mCEBOT by anchoring on pivot under the steering magnetic field. **e** Slipping motion trajectory and corresponding small space occupation. **f** Rolling motion trajectory and corresponding long step size. **g** Walking motion trajectory and corresponding biped locomotion characteristics. **h** Climbing motion trajectory and corresponding large height fluctuation. **i** mCEBOT unit in separated state slips across the narrow channel. **j** mCEBOT units assemble into bastinade shape to roll over the obstacle. **k** mCEBOT units assemble into biped structure to walk with two legs on wet surface like a human. **l** mCEBOT units assemble into hoe shape to climb by anchoring on the pivot.

reconfigured to bastinade shape, just 3 units are high enough to overcome the barrier (Fig. 4k, Supplementary Fig. 7b and Supplementary Movie 3). Due to the handicap from surface tension, wet surface is always a great challenge scenario for the locomotion of robot at small-scale (Supplementary Figure 7c). By reconfiguring into the biped structure, our mCEBOT is able to adopt the walking for locomotion. In this case, the resistance from surface tension can be greatly reduced benefited from the alternated point contact, which not only diminishes the required driving force but also prevents unintended separation, making mCEBOT work efficiently on wet surface (Fig. 4j, Supplementary Fig. 7d, and Supplementary Movie 3). For the hanging target that is higher than robot itself, except for high burst jumping or flying, it's almost impossible for a conventional robot to touch or manipulate the target. However, by rationally utilizing the pivot in the environment as a fulcrum, our mCEBOT can achieve this by configuring into hoe mode with climbing motion behavior (Fig. 4l and Supplementary Movie 3).

To further elaborate the environmental adaptability of mCEBOT, we set up an complexed environment containing wet surface (~300 μm water film), narrow slit (mezzanine with ~0.7 mm gap, ~58% of long unit's height), obstacle area (barrier with ~4.0 mm height, ~4.7 times of short unit's height) and hanging target (~3.5 mm height, ~1.1 times of the maximum height of mCEBOT), as shown in Fig. 5a, b. To cross the wet surface, these individual units are reconfigured to biped structure firstly. Under the action of spatially varying magnetic field (100 mT), the assembled biped mCEBOT moves forward ~30 mm in 70 s and finally step out of the scope of water coverage in 80 s by walking motion. To cross the narrow slit, the biped structure is firstly separated into individual units by decreasing the magnetic strength to 0 mT at 84 s. Then, driven by an oscillating magnetic field (100 mT, 1 Hz), these independent robot monomers slip

across the narrow mezzanine at 410 s by slipping motion. When meeting the high barriers, these individual units are reconfigured to bastinade shape by end-by-end assembling in 30 s (from 410 s to 440 s). As a result, under a rotating magnetic field (100 mT, 0.1 Hz), the assembled bastinade shaped mCEBOT overcomes the barrier at 452 s by rolling. In order to take the hanging target that is higher than robot itself height, the bastinade shaped mCEBOT is reassembled into hoe shape at 584 s. As the time-lapse images (Fig. 5c) and the height changing (Fig. 5d) shown, the target object is hanged on a vine-like branch with a ground clearance of ~3500 um which is ~1.1 times of the maximum height of bastinade shaped mCEBOT. Despite mCEBOT cannot jump or fly, by taking a 1500 um height vine-like branch as pivot, its reachable height can increase to ~4000 um (~125% of initial height) under the climbing motion. Since the enhanced reachable height already meets demand, the hanging target is successfully taken in 50 s. These results suggest that the environmental adaptability of robot at small-scale could be greatly enhanced through morphologies and behaviors reconfigurations (Supplementary Movie 3), offering great opportunities for tackling the tasks in harsh environments.

**Environment exploration and path marking.** Benefiting from the heterogeneous assembly endowed selective assembling and detaching of units, our mCEBOT can perform the tasks that conventional milli-scale robot cannot. For example, we can apply mCEBOT for unstructured environment exploration and path marking. On the one hand, mCEBOT can easily overcome complex terrain by the reconfiguration of morphologies and behaviors simultaneously. On the other hand, it can also realize the marking of valid paths by selective and stepwise unloading units, which can guide the locomotion of other robots and

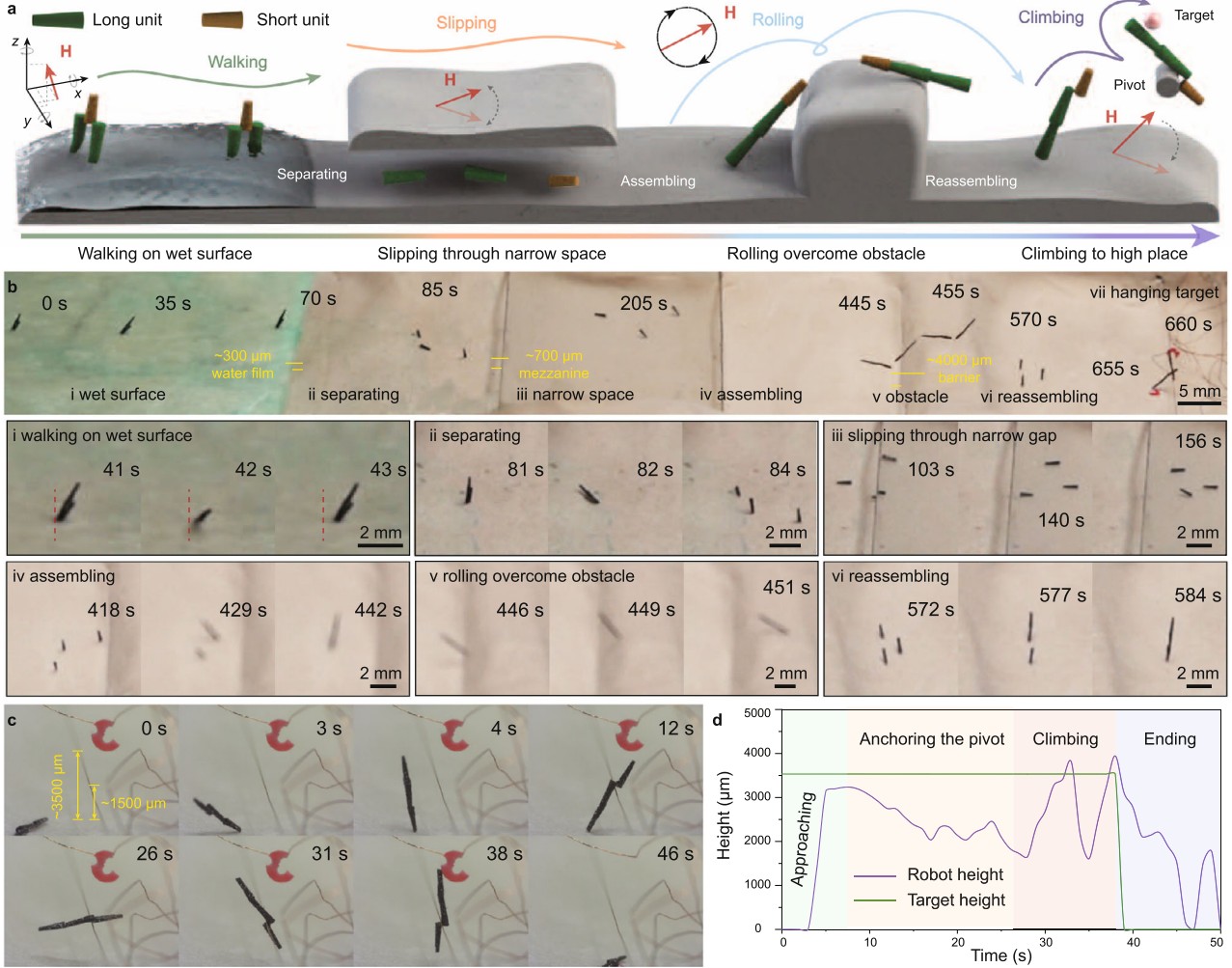

**Fig. 5 Combined multimodal locomotion modes of mCEBOT in a compound task. a** Schematic diagram of the complexed environment, where mCEBOT starts from the wet surface, and then needs to through a mezzanine and a barrier, finally has to take the hanging target. **b** Experimental results of mCEBOT in the compound task, mainly including walking on wet surface, separating into separated state, slipping through narrow space, assembling into bastinade shape, rolling overcome obstacle, reassembling into hoe shape and taking hanging target. **c** Time-lapse images of mCEBOT during hanging target taking. **d** Height changing of mCEBOT and target during the hanging target taking task.

improve their efficiency in passing through unstructured and complexed environments.

As shown in Fig. 6a, the environment explored as demo can be seen as a maze with complex terrain. During the mission, mCEBOT not only needs to overcome the complex terrain on the road (mainly exemplified by high barriers and narrow slits) but also choose the correct route among the many forks to reach the destination. In order to keep the environmental adaptability of mCEBOT and mark valid paths in detail, the initial number of units of mCEBOT needs to be determined according to the complexity of the environment and tasks. On the one hand, too few initial units will limit the reconfiguration of morphologies and behaviors, and may not have enough marks for path marking. On the other hand, the number of units is not the more the better because the number of units far exceeding the demand will cause a burden on the control and actuation. After evaluation, the initial number of units in our demonstration is set as 5, and one of the units is left at the starting point as the first marker. When a high barrier is encountered, the remaining units will assemble into bastinade shape and quickly overcome it by rolling motion (Fig. 6b). While for a narrow slit, the units will remain in separated state and traversing it by slipping motion (Fig. 6c). Benefiting from

the selective assembly and the use of locally actuation magnetic field, mCEBOT can unload partial units as markers at desired sites. For example, mCEBOT can separate a unit at the fork or before the obstacle. If the route is correct or the obstacle is successfully surmountable, then the detached unit will be preserved as a mark, otherwise the detached unit will be recycled when mCEBOT returns. Based on this strategy, our mCEBOT finally passed 4 obstacle areas (2 high barriers, 2 narrow slits) within 450 s and left 5 markers (3 for forks, 1 for starting point, one for destination) during the exploration process (Supplementary Movie 4). To verify the effectiveness of environment exploration and path marking, we apply a new unit to conduct path following and marks recycling. As shown in Fig. 6d, the new unit can successfully reach the destination within 256 s under the guidance of the markers. In addition to the guiding role, the marks can also be recycled to assemble with new unit to enhance its environmental adaptability. Figure 6e, f is the motion trajectories of each unit of mCEBOT in the process of path marking and path following. By comparison, it can be found that under the guidance of markers, the robot can omit the attempt of invalid paths when passing through the same unstructured environment, and reach the destination faster and more efficiently.

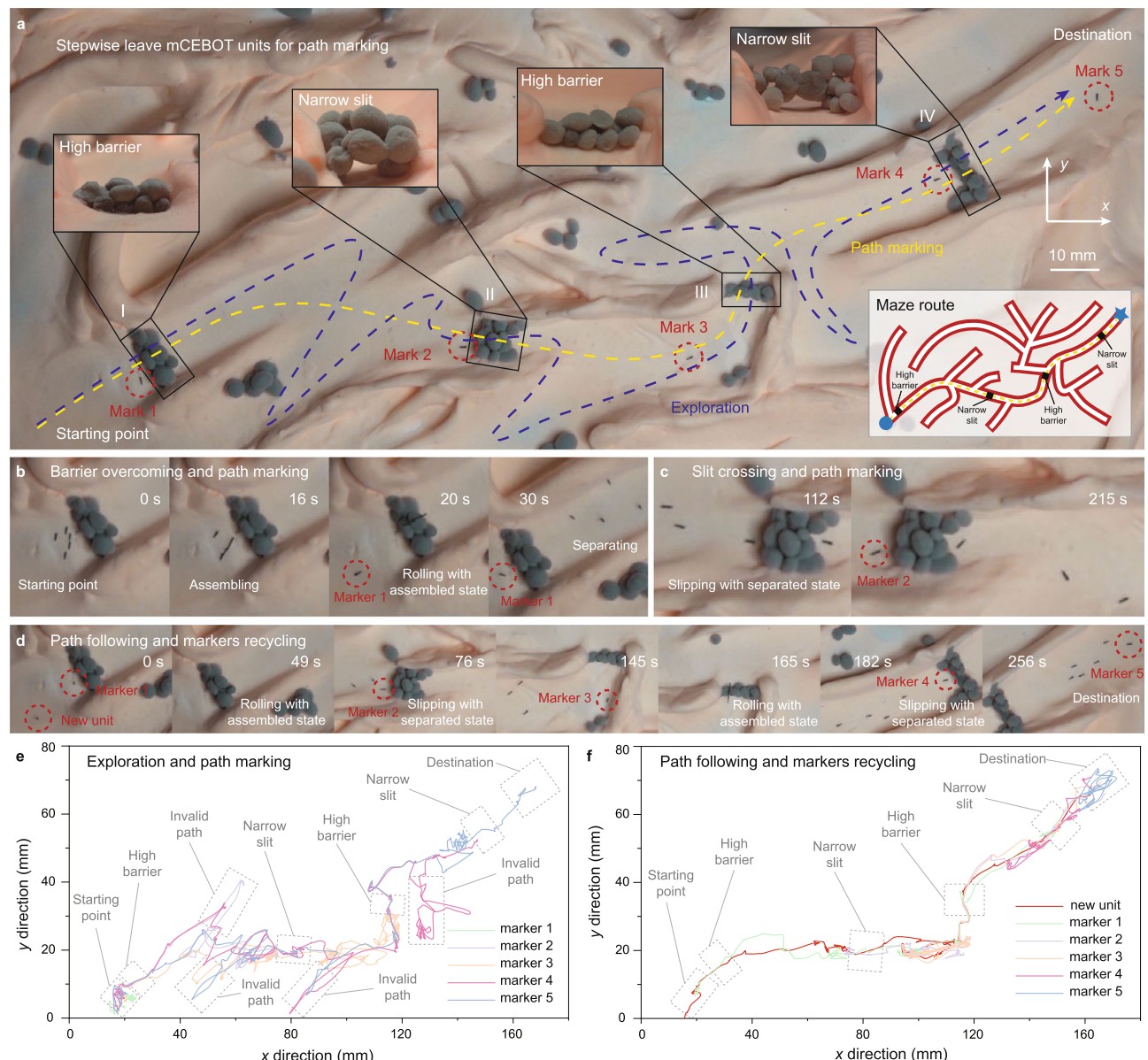

**Fig. 6 Demonstration of mCEBOT for environment exploration and path marking. a** Schematic diagram of maze route with complex terrain. **b** Barrier overcoming by assembling units into rolling mCEBOT, and corresponding path marking during exploration. **c** Slit crossing by units slipping with separated state, and corresponding path marking during exploration. **d** Path following of new unit, and corresponding markers recycling. **e** The trajectory of mCEBOT units during environment exploration and path marking. **f** The trajectory of mCEBOT units during path following and markers recycling.

## Discussion

Small robots can effectively perform certain task if the architecture is design properly. However, limited to their inherent physical architecture, it's difficult for them to adapt to the unstructured environment and complete complexed tasks. The reconfigurable modular robot is a long-standing goal of scientists, which can express diverse morphologies to adapt to unstructured environments and switch multimodal motion behaviors according to different tasks. Yet, considering the limitation in the conscious assembly and on-demand separation at small-scale, it remains a grand challenge for the construction of milli-sclae modular robots. The magnetic field that allows non-contact interaction, lenient position errors, and no mechanical connection has been adopted in modular robot assembly[29–31], and the facile magnetization and demagnetization of soft magnetic materials show potential in the assembly and separation of the modular robot at small-scale. However, considering the uncertain units number, arbitrary location distribution and inevitable magnetic field singularity, the differentiated control of designated units for conscious assembly is still difficult under the global action of magnetic field. Moreover, for the obtained new assembled unit, the sustainability of assembly and actuation should be considered. Besides that, the lack of diverse behaviors and functions also limits the potential application of modular robots at small-scale.

By introducing heterogenous assembly, mCEBOT can achieve multiple morphologies and versatile behaviors simultaneously. First, the different dimensions of short and long units lead to different magnetic responses, which make up for the undifferentiated actuation of the global magnetic field and provide the possibility for conscious assembly. As the demonstration shown, mCEBOT units can be manipulated with different trajectories for connection even under the same magnetic field. Secondly, the multiplex assembly (including end-by-end and side-by-side connection) can be further achieved for morphologies and behaviors

enriching, which greatly enhance the environmental adaptability of mCEBOT. Thirdly, the selective assembling and detaching of units make mCEBOT can on demand leave some units for complex tasks, such as the active retention of mCEBOT's units during the environment exploration for path marking, the localized separation of mCEBOT's units (with cargo inside) in different target areas for multi-regional delivery, the stepwise degradation of mCEBOT's units (with different drugs inside) in GI tract for combination therapy. Besides heterogenous assembly, the heterogeneity in materials (e.g., response to pH, temperature, humidity, and light) and shapes (e.g., delicate structures by 3D printing) is also extensible to endow the robot design with more possibilities in the future work. The concept of mCEBOT will provide a prospective strategy for robot constructions and functions enabling, which will benefit a wide spectrum of applications at small-scale.

## Methods

**Raw materials of heterogeneous units**. Eudragit polymers (L100-55) are provided by Evonik Corporation (shanghai, CN). Iron micro powder (Spherical, APS 6–10 micron, reduced, 99.5%) is obtained from Alfa Aesar. Ethanol with an assay of 99.9% is used for all solution preparations, which is purchased from Anaqua Global International Inc. Limited. At first, the 30% wt Eudragit polymers solution is prepared by dissolving Eudragit powder in pure ethanol following the stirring at 500 rpm for 24 h with the magnetic stirrer (RCT basic, German IKA Corporation). Then the raw material of mCEBOT units is obtained by mixing the prepared Eudragit polymers solution and 30% iron powder. All the chemicals are with the purity of analytical reagent grade used as received without further purification.

**Magnetic moment density measurement**. The magnetic moment density of units is detected by vibrating sample magnetometer (VSM600, Quantum Design). First, the units with 30% magnetic particle mass fraction are prepared according to the mentioned magnetic-field-assisted manufacturing. Then, transfer them into VSM after weighing. The applied magnetic field with an interval of 50 mT, first increases from 0 to 2000 mT and then decreases to −2000 mT and finally back to 0 mT. During which, the generated magnetic moment of units under different magnetic field is recorded, and its magnetic moment density is calculated after dividing by the total weight.

**Unit dimensions under different fabrication conditions**. To investigate the effect factors of unit dimensions, including height, bottom radius, and bottom angle, we conduct the fabrication of mCEBOT units under different conditions. Here, the top substrate is removed for obtaining the maximum growth height, the magnetic particle mass of raw material changes from 10 to 40%, and the applied magnetic field strength increases from 40 to 120 mT. After recording all units' dimensions under different fabrication conditions, we classify and analyze the corresponding results by the control variable methods. During which, the maximum growth height of unit is proportional to both the magnetic particle mass and applied magnetic field strength. While the bottom radius is mainly dependent on the magnetic particle mass but has no significant relationship with the applied magnetic field. On the contrary, the bottom angle increases as the applied magnetic field strength increases but expresses no significant relationship with the magnetic particle mass.

**Bonding force measurement**. The bonding force of the bistable assembly, including end-by-end and side-by-side connection, is measured by manual separating corresponding units under magnetization state. Here, two same long units are adopted to demonstrate the end-by-end and side-by-side connection, and the applied magnetic field changes from 50 to 100 mT. To separate the assembled units effectively and record the bonding force precisely, one unit is fixed on the force measurement platform (Sartorius entris analytical balance, resolution 0.1 mg) and the other unit is pulled away by the micromanipulator (HOURS, OSMS20-XYZ) with a non-magnetic fixture. During which, the pulling direction is always particular to both the connection surface and force measurement platform, i.e., along the axial direction in end-by-end connection and along the radial direction in side-by-side connection.

## Data availability

All data needed to evaluate the conclusions in the paper are present in the paper and the supplementary materials. Additional data related to this paper may be requested from the authors.

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

## Acknowledgements

This work was supported by National Science Foundation of China under Grants 61922093 (Y.S.) and U1813211 (Y.S.), Hong Kong RGC General Research Fund under Grants CityU 11216421 (Y.S.), Shenzhen-Hong Kong-Macau Science and Technology Project under Grants SGDX20201103093003017 (Y.S.), and ShenZhen Key Basic Research Project under Grants JCYJ20200109114827177 (Y.S.).

## Author contributions

X.Y. designed the robot, carried out the experiments, analyzed the results, and developed the robot model. R.T. carried out the experiment in material synthesis. H.L. contributed to scientific discussion and experiment design. T.F. conceived the idea and provided valuable comments for the project implementation. Y.S. conceived the idea and led the project.

## Competing interests

The authors declare no competing interests.
