## [Peer Review File · Nature Communications]

Milli-scale cellular robots that can reconfigure morphologies
and behaviors simultaneouslyReviewers' comments:

Reviewer #1 (Remarks to the Author):

The authors report on a modular microrobotic design that can reconfigure into different shapes. They claim that the robot is decomposable in certain pH condition due to heterogeneity of the robotic design. The work lacks novelty especially with regard to the conceptualization and conclusion. Besides, the method used in this study is lacking and is not reproducible. I do not recommend the publication of this work in Nature Communication.

Other concerns related to the work of the authors are:

- 1) The terminologies used such as "master and slave" is misleading and inappropriate and should be avoided. End-to-end or side-by-side assembly is suitable term to use.
- 2) All locomotion modalities have been shown before in a single unit robot unit in the cited reference [Hu, W., Lum, G. Z., Mastrangeli, M., & Sitti, M. (2018). Small-scale soft-bodied robot with multimodal locomotion. Nature, 554(7690), 81-85.]
- 3) The heterogeneous robot assembly in the authors work is not novel. [Check the work of Li Zhang and colleagues: Ultra-extensible ribbon-like magnetic microswarm, <https://www.nature.com/articles/s41467-018-05749-6>]
- 4) Surface tension effect on robot fabrication lacks clarity.
- 5) In wet condition, the author did not show which mechanism applies to the locomotion governing the walking principle of the robot. Is friction important in understanding the contact point or the buoyancy?
- 6) For realistic hyperthermia, the decomposition of robot is to be shown in liquid environment. Thus, the reported temperature value is ambiguous.
- 7) Biocompatibility of the polymer material for the robot is a big concern. Are the magnetic particles embedded inside or are coated on the polymer surface?
- 8) I suggest the authors should provide in vivo experiment to show the potential application of such robot.

Reviewer #2 (Remarks to the Author):

This work presents a strategy to construct reconfigurable cellular robot (CEBOT) at the small scale. The author introduced the idea of "heterogeneous" into the design of the micro unit, and achieved several types of morphologies (independent monomer, bastinade shape, biped structure) and adaptive mobility (slipping, rolling, walking) in unstructured environments by simple magnetic control. The author demonstrated the . Secondly, the author explored some biocompatible environment adaptive materials for the CEBOT design, which endows CEBOT functions for implementing biomedical tasks, as the demonstrated local hyperthermia and multi-region releasing. I like the idea of "heterogeneous" and "functional" proposed in this manuscript. It provides a prospective strategy for micro robot design in both modularization and functionalization, and makes a significant step towards the realization of intelligent modular small robot. Yet, before publication, the manuscript should be improved by adopting multilevel methodological approaches and providing more details.

Major issues:

1. Fabrication: The paragraph about unit fabrication requires elaboration to clarify the adequate composite constitution, together with generatable magnetic force and torque. Actual generatable force

and torque are missing and it should be calculated with a normalized sample with real values.

2. Assembly: The controllable assembly and separation of heterogeneous micro units are fascinating for me. However, I have some concerns about the effect of invalid interaction between non-selected units on the assembly process of selected units, such as the unexpected assembly of non-selected units due to the global actuation of magnetic field. Authors are suggested to discuss these effects.

3. Assembly: If the surface is uneven, such as a ditch or a bulge, can the proposed assembly strategy still be valid? How does the magnetic-assist assembling adapt to these harsh conditions and potential position errors?

4. Characterization: In Fig. 2D, the authors detect the bonding force between two robotic units to verify the stability of magnetic-assist assembling. I found the bonding force of end by end is along the long axis of the unit, while the bonding force of side by side is along the radial direction of the unit. Please explain why.

5. Locomotion: Authors claim that the locomotion of micro robots on wet surfaces is difficult due to the surface tension, that's also the meaning of constructing the walking robot. For some biomedical applications that wet surfaces are everywhere. Does the actuation and assembly of micro units remain effective on wet surfaces?

6. Function: Despite the demonstration of adaptive multi-region releasing of CEBOT, the discussion of its potential application is missing. I believe this programmable and selective releasing by assembling the units with different decomposition properties could be an advantage of heterogeneous modular robot.

Minor issues:

1. On the decomposition of robotic units, are there any other methods instead of applying the magnetic stirring?

2. Besides the environmental pH and materials, the environmental temperature will also influence the decomposition process. The detailed environmental conditions should be provided in the investigation of decomposition.

3. Authors are suggested to discuss the differences between the assembly of single CEBOT and multiple CEBOTs in a large number of robotic units.

4. What kind of Iron micro powder is used? Reference to biocompatibility is necessary especially when discussing its dissolution in biomedical applications.

5. Details on what bars, box or whiskers represent should be added in figure legends.

6. The grammar and style of writing have some errors throughout the manuscript and need to be checked and corrected. This will improve the work greatly and will make it more readable. For instance,

a. "Benefit from" at the beginning of the sentence should be "Benefiting from";

b. "different height" should be "different heights";

c. "separates on-demand (at 123.5 s)" should be "separate on-demand (at 123.5 s)";

d. "After that, we separate it into individual units by decreasing magnetic field strength to 0 mT, and accelerate the decomposition by applying magnetic stirring (50mT, 120 r/min)." a comma is missing;

e. "the architecture is design properly" should be "the architecture is designed properly".

Reviewer #3 (Remarks to the Author):

This paper some interesting points and design tricks, such as a hard magnetic to bond soft magnetic modules as a master-slave system. However, there are many similar works, and the authors should clearly define the real progress of their work.

A few magnetic modules can arrange with different shapes, which is not surprising. The demonstrations in Fig. 3 and 4 only show 3 modules, so the method does not appear to be scalable. The authors claim that their robotic system has advantages for navigating in unconstructed environments, and "the unstructured environment" in (Fig 4) is a well designed wet surface, a narrow space, and around an obstacle with fined tuned manual control and visual feedback.

I wonder what will happen if it is put it in a real intestine, and what level of adaptability the system will show.

Finally, is it overall better than the soft robot with multimodal locomotion (Metin Sitti's Nature paper)? I am afraid not, as that robot can jump.

Reviewer#1:

The authors report on a modular microrobotic design that can reconfigure into different shapes. They claim that the robot is decomposable in certain pH condition due to heterogeneity of the robotic design. The work lacks novelty especially with regard to the conceptualization and conclusion. Besides, the method used in this study is lacking and is not reproducible. I do not recommend the publication of this work in Nature Communication.

Response: We regret we didn't present the novelty and methodologies of our work clearly. In this revised version, we have substantially revised our manuscript by clarifying the contribution, deleting the overclaimed contexts, adding more experiment data to support the statement, and reorganized/rewrote the paper. Please check our responses to your comments item by item:

1. The terminologies used such as "master and slave" is misleading and inappropriate and should be avoided. End-to-end or side-by-side assembly is suitable term to use.

Response: Thanks for your comments and suggestion. In the revised version, we have removed the terminologies "master and slave", and used a direct description, that "long units" and "short units", to distinguish the two types of units with different dimensions.

2. All locomotion modalities have been shown before in a single unit robot unit in the cited reference [Hu, W., Lum, G. Z., Mastrangeli, M., & Sitti, M. (2018). Small-scale soft-bodied robot with multimodal locomotion. *Nature*, 554(7690), 81-85.]

Response: Modular robot (CEBOT) and single unit robot are two different kinds of robot construction strategies. Despite they may show similar locomotion, they are fundamentally different. For instance, compared with the single unit robot, not only the fantastic robot mentioned by you, CEBOT has several unique properties due to its separatable and reorganizable structure:

- 1) The CEBOT could achieve richer morphologies by the heterogeneous assembling and separating of modular units. That is different from the single unit robot which can only perform simple bending or stretching deformation restricted by the inherent physical structure.
- 2) The CEBOT could have greater adaptabilities with large-scale spanned. For example, it can easily cross the slit with the smallest unit size by separating, and overcome the obstacles more than ten times the size of the unit by assembling.
- 3) The modularity design of CEBOT could bring new possibilities to perform more complex tasks, such as modular replacement for self-healing, partial separation and recycle for navigation, stepwise degradation and release for multi-point cargo delivery, etc.

In this revised version, we have added more data to demonstrate the unique advantages of modular mCEBOT, such as in complexed environment task and environment exploration. Please check main text page 13-14.

3. The heterogeneous robot assembly in the authors work is not novel. [Check the work of Li Zhang and colleagues: Ultra-extensible ribbon-like magnetic microswarm, <https://www.nature.com/articles/s41467-018-05749-6>]

Response: Thanks for listing this reference. We regret that we didn't present our work clearly in previous version. In fact, the separating-assembling characteristics of CEBOT is quite different from the virtual-assembling of magnetic swarm robots [1-4]:

- 1) The separating-assembling strategy of CEBOT can achieve the selective actuation, control and assembling of independent units, which endows the selective and bistable assembling between modular units. While for the virtual assembling of magnetic swarm robots, it is based on indiscriminate reunion with similar virtual connecting.
- 2) The separating-assembling strategy of CEBOT enable it more motion behaviors (slipping, walking, rolling, climbing) and application scenarios (liquid, wet surface, dry land). For comparison, the motion of magnetic swarm robots are usually confined to the liquid environment due to the virtual connection between units.
- 3) Swarm robots mainly focus on the collective behavior of massive number of units while ignoring the behavior and properties of individual units. For modular CEBOT, the selective detaching of individual unit can be used for path marking, the stepwise degradation of individual unit can achieve multi-point release, and the adoption of different functional materials for different units' fabrication can enable the fusion of rich functions.

In short, we think both swarm robots and modular CEBOT have their unique properties and application areas. We didn't say CEBOT is better than swarm robot (actually, swarm robot is very amazing), but would like to focus on the modular robot at millimeter scale, which is also important but rarely addressed. To present it clearly, we have defined the real progress of our work by comparing with other similar works, and revised the manuscript with the most seriousness. Please check main text page 2-3.

Modifying:

Introduction, main text page 2-3:

Table 1. The comparison of existing assembly methods for small-scale modular robot

	Independent control units	Reversibility of assembly	Morphologies reconfiguration	Behaviors reconfiguration	Features	References
Self-assembly	-	Poor	Poor	Poor	Assembly process is uncontrollable	[5,6]
Irreversible assembly	-	-	-	Poor	Non-reversing	[7,8]
Virtual assembly	-	Good	Good	Good	Global actuation and limited to liquid environment	[1-3]
Assisted assembly	Good	Poor	Good	Poor	Less diversity in behaviors and functions	[9-14]
Heterogeneous assembled mCEBOT	Good	Good	Good	Good	Reconfigure morphologies and behaviors simultaneously	This work

1. “-” means cannot achieve; “Poor” means few can achieve; “Good” means most can achieve.

However, under the critical constraints of space and energy consumption, constructing modular robot at small-scale is still very difficult limited by the lack of conscious approaching, diverse docking, and selective detaching strategies between micro units (Table 1). Although swarm design can cluster massive micro/nano particles to perform certain collective behaviors [1-3], drawbacks in poor monomer control, non-selective unit approaching, and virtual modular connection make it only works under the global magnetic actuation. Moreover, despite the diverse achievable patterns of swarm, its

motion behaviors are limited and mostly confined to the liquid environment. Applying additional frameworks, restriction or assistance [9-14] can also collect or assemble small units as desired, however these strategies impose more or less restrictions on the reversibility of assembly and the application of robot. Furthermore, most of these strategies only focus on the morphological reconfiguration and ignore the diversity of behaviors and functions of robot, which makes the small-scale modular robot far from reliable in practice.

4. Surface tension effect on robot fabrication lacks clarity.

Response: Thanks for your suggestion. The fabrication method of our mCEBOT is developed based on the existed and mature magnetic-assist-manufacturing [15]. Although this method is less accurate and repeatable than mold manufacturing, it has better processing flexibility and there is no need to consider the demolding problem at small size. In addition, the influences or effects of magnetic field strength, solution viscosity and surface tension to the fabrication have been introduced in the previous works [16-18]. For readers to better understand the role of surface tension in the fabrication, we provide the detailed mechanical analysis in the supplementary materials and Fig. S2B (Fig. R1-1).

Fig. R1-1. The formation of mCEBOT unit under the modified magnetic-assist-manufacturing. As the mechanical analysis shown the frustum shape is achieved under the combined action of magnetic field, surface tension and gravity. During which, the surface tension is related to the maximum achievable height and taper of mCEBOT unit. Under the same magnetic field, the larger surface tension will lead to larger taper and smaller achievable height of mCEBOT unit.

5. In wet condition, the author did not show which mechanism applies to the locomotion governing the walking principle of the robot. Is friction important in understanding the contact point or the buoyancy?

Response: We thank the referee very much for his/her valuable comments. During walking, the mCEBOT uses one of its legs to contact with the ground and takes the contact point as an anchor point. By rotating and tilting its body under the action of a magnetic field, the distance between the two legs is converted into an effective step length. Then, continuous walking is achieved by alternating legs in contact with the ground as anchor points. Similar to human bipedal walking, friction is important to ensure the stability of the contact point as an anchor point. If there is no friction or the friction is too small, the contact point will slip during the stepping, causing the mCEBOT to swing in place. In order to help readers to understand the walking process better, we have provided detailed force and gait analysis in supplementary materials.

Fig. R1-2. Analysis of walking. **A**, The mechanical analysis of walking under the action of spatially varying magnetic field. **B**, Gait analysis of walking, which converts the distance between the two legs into effective step length.

6. For realistic hyperthermia, the decomposition of robot is to be shown in liquid environment. Thus, the reported temperature value is ambiguous.

Response: Thanks for your comments. Considering it's impossible to obtain the exact temperature in realistic hyperthermia due to the influences from liquid environment, we have withdrawn the hyperthermia demonstration and replacing it with more representative engineering application task. Please check main text page 13-14.

Modifying:

Locomotion adaptability in complexed environment, main text page 13:

To further elaborate the environmental adaptability of mCEBOT, we set up a complexed environment containing wet surface (~300 μm water film), narrow slit (mezzanine with ~0.7 mm gap, ~58% of long unit's height), obstacle area (barrier with ~4.0 mm height, ~4.7 times of short unit's height) and hanging target (~3.5 mm height, ~1.1 times of the maximum height of mCEBOT), as shown in Fig. 5A and Fig. 5B. To cross the wet surface, these individual units are reconfigured to biped structure firstly. Under the action of spatially varying magnetic field (100 mT), the assembled biped mCEBOT moves forward ~30 mm in 70 s and finally step out of the scope of water coverage in 80 s by walking motion. To cross the narrow slit, the biped structure is firstly separated into individual units by decreasing the magnetic strength to 0 mT at 84 s. Then, driven by an oscillating magnetic field (100 mT, 1 Hz), these independent robot monomers slip across the narrow mezzanine at 410 s by slipping motion. When meeting the high barriers, these individual units are reconfigured to bastinade shape by end-by-end assembling in 30 s (from 410 s to 440 s). As a result, under a rotating magnetic field (100 mT, 0.1 Hz), the assembled bastinade shaped mCEBOT overcomes the barrier at 452 s by rolling. In order to take the hanging target that is higher than robot itself height, the bastinade shaped mCEBOT is reassembled into hoe shape at 584 s. As the time-lapse images (Fig. 4C) and the height changing (Fig. 4D) shown, the target object is hanged on a vine-like branch with a ground clearance of ~3500 μm which is ~1.1 times of the maximum height of bastinade shaped mCEBOT. Despite mCEBOT cannot jump or fly, by taking a 1500 μm height vine-like branch as pivot, its reachable height can increase to ~4000 μm (~125% of initial height) under the climbing motion. Since the enhanced reachable height already meets demand, the hanging target is successfully taken in 50 s. These results suggest that the environmental adaptability of robot at small-scale could be greatly enhanced through morphologies and behaviors reconfigurations (movie S3), offering great opportunities for tackling the tasks in harsh environments.

Fig. 4. Combined multimodal locomotion modes of mCEBOT in a compound task. **A**, Schematic diagram of the complexed environment, where mCEBOT starts from the wet surface, and then needs to through a mezzanine and a barrier, finally has to take the hanging target. **B**, Experimental results of mCEBOT in the compound task, mainly including walking on wet surface, separating into separated state, slipping through narrow space, assembling into bastinade shape, rolling overcome obstacle, reassembling into hoe shape and taking hanging target. **C**, Time-lapse images of mCEBOT during hanging target taking. **D**, Height changing of mCEBOT and target during the hanging target taking task.

mCEBOT for environment exploration and path marking, main text page 14:

Benefiting from the heterogeneous assembly endowed selective assembling and detaching of units, our mCEBOT can perform the tasks that conventional milli-scale robot cannot. For example, we can apply mCEBOT for unstructured environment exploration and path marking. On the one hand, mCEBOT can easily overcome complex terrain by the reconfiguration of morphologies and behaviors simultaneously. On the other hand, it can also realize the marking of valid paths by selective and stepwise unloading units, which can guidance the locomotion of other robots and improve their efficiency in passing through unstructured and complexed environments.

As shown in Fig. 5A, the environment explored as demo can be seen as a maze with complex terrain. During the mission, mCEBOT not only needs to overcome the complex terrain on the road (mainly exemplified by high barriers and narrow slits), but also needs to choose the correct route among the many forks to reach the destination. In order to keep the environmental adaptability of mCEBOT and mark valid paths in detail, the initial number of units of mCEBOT needs to be determined according to the complexity of the environment and tasks. On the one hand, too few initial units will limit the reconfiguration of morphologies and behaviors, and may not have enough marks for path marking. On the other hand, the number of units is not the more the better because the number of units far exceeding the demand will cause a burden on the control and actuation. After evaluation, the initial number of units in our demonstration is set as 5, and one of the units is left at

the starting point as the first marker. When a high barrier is encountered, the remaining units will assemble into bastinade shape and quickly overcome it by rolling motion (Fig. 5B). While for a narrow slit, the units will remain separated state and traversing it by slipping motion (Fig. 5C). Benefiting from the selective assembly and the use of locally actuation magnetic field, mCEBOT can unload partial units as markers at desired sites. For example, mCEBOT can separate a unit at the fork or before the obstacle. If the route is correct or the obstacle is successfully surmountable, then the detached unit will be preserved as a mark, otherwise the detached unit will be recycled when mCEBOT returns. Based on this strategy, our mCEBOT finally passed 4 obstacle areas (2 high barriers, 2 narrow slits) within 450 s and left 5 markers (3 for forks, 1 for starting point, one for destination) during the exploration process. To verify the effectiveness of environment exploration and path marking, we apply a new unit to conduct path following and marks recycling. As shown in Fig. 5D, the new unit can successfully reach the destination within 256 s under the guidance of the markers. In addition to the guiding role, the marks can also be recycled to assemble with new unit to enhance its environmental adaptability. Fig. 5E and 5F are the motion trajectories of each unit of mCEBOT in the process of path marking and path following. By comparison, it can be found that under the guidance of markers, the robot can omit the attempt of invalid paths when passing through the same unstructured environment, and reach the destination faster and more efficiently.

Fig. 5. Demonstration of mCEBOT for environment exploration and path marking. A, Schematic diagram of maze route with complex terrain. B, Barrier overcoming by assembling units into rolling mCEBOT, and corresponding path marking during exploration. C, Slit crossing by units slipping with separated state, and corresponding path marking during exploration. D, Path following of new unit, and corresponding markers recycling. E, The trajectory of mCEBOT units during environment exploration and path marking. F, The trajectory of mCEBOT units during path following and markers recycling.

7. Biocompatibility of the polymer material for the robot is a big concern. Are the magnetic particles embedded inside or are coated on the polymer surface?

Response: In our mCEBOT, the magnetic particles are fully mixed with the polymer material rather than just coated on the surface. The adopted Eudragit polymer has been widely used in the biomedical field, and its biocompatibility has been fully proved [19-21]. For the magnetic particles, our choice is the iron micro powder. Since Fe is a natural metal (ferritin) in human body, we can metabolize iron particles into our elements [22]. Iron is safe for the individual within a moderate range and more biocompatible than other magnetic particles including Ni as well as Co [23]. By now, iron and its oxides have also been widely used in biomedical applications, such as magnetic microrobot for drug delivery [24,25] and peripheral stent in tissue engineering [26].

8. I suggest the authors should provide in vivo experiment to show the potential application of such robot.

Response: We fully agree that the in vivo animal test is a key step to prompt the robot to biomedical engineering. However, limited by existing in vivo imaging and actuation strategies, there are no reconfigurable magnetic microrobots for animal or human experiments [1-14]. This work focuses on the construction of modular magnetic robot at millimeter scale by heterogeneous assembling, and its potential advantages and applications endowed by the reconfigurable morphologies and motion behaviors. To avoid the overclaim, we have changed the application demonstration of in vitro bioengineering into engineering area based on the results obtained so far, including the taking hanging targets (main text page 13, Fig. 4) and the exploration of unstructured environments (main text page 14, Fig. 5).

Reviewer#2:

This work presents a strategy to construct reconfigurable cellular robot (CEBOT) at the small scale. The author introduced the idea of “heterogeneous” into the design of the micro unit, and achieved several types of morphologies (independent monomer, bastinade shape, biped structure) and adaptive mobility (slipping, rolling, walking) in unstructured environments by simple magnetic control. Secondly, the author explored some biocompatible environment adaptive materials for the CEBOT design, which endows CEBOT functions for implementing biomedical tasks, as the demonstrated local hyperthermia and multi-region releasing. I like the idea of “heterogeneous” and “functional” proposed in this manuscript. It provides a prospective strategy for micro robot design in both modularization and functionalization, and makes a significant step towards the realization of intelligent modular small robot. Yet, before publication, the manuscript should be improved by adopting multilevel methodological approaches and providing more details.

Response: We thank the referee very much for the review and high recognition of our work. We have revised our manuscript based on the referee’s comments and suggestions, please check the following response item by item.

1. Fabrication: The paragraph about unit fabrication requires elaboration to clarify the adequate composite constitution, together with generatable magnetic force and torque. Actual generatable force and torque are missing and it should be calculated with a normalized sample with real values.

Response: Thanks for your comments. In the revised version, we have clarified the materials and corresponding preparation in the “Materials and Methods” section. Further, the generatable magnetic force and torque are analyzed with model and calculated with a normalized sample with real values. Please check in supplementary materials page 3-4.

Modifying:

Analysis of the actuation ability of mCEBOT units, supplementary materials page 3-4
The mCEBOT unit will experience magnetic force and torque under the external magnetic field. Assuming the single magnetic particle is a point-like dipole with a magnetic susceptibility χ under the magnetic field B , R is the diameter of the spherical magnetic particle. For a specific mCEBOT unit, N is the total contained magnetic particles. Then, the endured magnetic force of mCEBOT unit can be expressed as [27]: $F = N \cdot \frac{\pi R^3}{6} \cdot \chi \cdot \nabla B$.

During which, the total number of magnetic particles N and the external magnetic field gradient ∇B are adjustable, which determine the magnetic force, i.e. the magnetic force of mCEBOT unit is proportional to the magnetic particle mass fraction and applied magnetic field. Since the magnetic particles inside the unit are aligned into magnetic chain during the curing process, the mCEBOT unit will endure magnetic torque until the magnetic field consists with the direction of magnetic chain. Assuming the magnetic chains are distributed evenly, d is the distance between inter particles, n is the number of the magnetic particles in a single chain, α is the angle between the direction of magnetic field and magnetic chain. Then, the total magnetic torque of the mCEBOT unit can be

expressed as: $T = N \frac{4\mu_0 n \chi^2 R^6 \pi}{3d^3} B^2 \sin 2\alpha$. For a specific unit, n is proportional to N , and

d is proportional to $\frac{1}{N}$, i.e. the magnetic torque of mCEBOT unit is proportional to the magnetic particle mass fraction, applied magnetic field B and the phase angle α .

The calculation of actual generated magnetic force and torque

Since the magnetization of mCEBOT unit can be measured by the VSM and its volume can be estimated according to the shape as well as size, the generated magnetic force and torque can be calculated. For the mCEBOT unit with 30% magnetic particle mass fraction, its magnetization M under 100 mT magnetic field is about 230.8 KA/m. The volume of mCEBOT unit can be expressed as:

$$V = \frac{1}{3} \pi \left[R^3 \tan \alpha - \frac{(R \tan \alpha - L)^3}{\tan^2 \alpha} \right] \quad (1)$$

where L is the body length of unit (850 μm for short unit and 1200 μm for long unit), $R=200 \mu\text{m}$ is the bottom radius, $\alpha=83^\circ$ is the bottom angle. Then, the maximum generated magnetic force and torque of long unit and short unit under the magnetic field (strength 100 mT and gradient 4750 mT/m) can be calculated:

$$F_{short} = V_1 \cdot M \cdot \nabla B = 6.6 \times 10^{-2} \text{ mN} \quad (2)$$

$$T_{short} = V_1 \cdot M \cdot B \cdot \sin\left(\frac{\pi}{2}\right) = 1.4 \times 10^{-6} \text{ N} \cdot \text{m} \quad (3)$$

$$F_{long} = V_2 \cdot M \cdot \nabla B = 7.2 \times 10^{-2} \text{ mN} \quad (4)$$

$$T_{long} = V_2 \cdot M \cdot B \cdot \sin\left(\frac{\pi}{2}\right) = 1.5 \times 10^{-6} \text{ N} \cdot \text{m} \quad (5)$$

2. Assembly: The controllable assembly and separation of heterogeneous micro units are fascinating for me. However, I have some concerns about the effect of invalid interaction between non-selected units on the assembly process of selected units, such as the unexpected assembly of non-selected units due to the global actuation of magnetic field. Authors are suggested to discuss these effects.

Response: That's a very interesting and insightful question. Due to the global actuation of the magnetic field, it is difficult to control the selected units while without affecting others. As reviewer mentioned, the unexpected assembling of non-selected units does have a chance of happening, especially when a large number of units are densely distributed. In the case of redundant units numbers, the unexpected assembling of a small number of non-selected units does not affect the final construction of modular robot at all. But when there is no redundancy in the number of units, unexpected assembling of non-selected units will reduce the number of assemblable units. In this case, it is necessary to use terrain or obstacles to separate the unexpected assembling, or to remove the magnetic field to separate all units for reassembling. Fortunately, the number of units currently adopted (<10) is far from sufficient to form a dense distribution, so the probability of unexpected assembling of non-selected units is low. In one word, the unexpected assembling of non-selected units has limited impact on the assembling process of selected units, and can be resolved even happened.

3. Assembly: If the surface is uneven, such as a ditch or a bulge, can the proposed assembly strategy still be valid? How does the magnetic-assist assembling adapt to these harsh conditions and potential position errors?

Response: We thank the referee very much for his/her constructive comments. The unevenness of the terrain mainly affects the locomotion of the unit, which will lead to deviations between the actual path and the preset path, and affect the efficiency of assembly. Benefiting from the non-contact triggering and no specific connection mechanism, the magnetic attraction based assembly between units can tolerate large position errors. As shown in Fig. 2B, under the action of magnetic field, the selected units can achieve effective assembly as long as their relative position meets a certain threshold. In addition, when the harsh terrain is only in specific areas, it can be circumvented by adjusting the assembly paths. For the harsh terrain that is unavoidable, path and position errors can be compensated during subsequent movements. We have added these in the revised version, please check main text page 6 and supplementary materials page 7.

4. Characterization: In Fig. 2D, the authors detect the bonding force between two robotic units to verify the stability of magnetic-assist assembling. I found the bonding force of end by end is along the long axis of the unit, while the bonding force of side by side is along the radial direction of the unit. Please explain why.

Response: For the end-to-end assembling, the binding force along the long axis mainly depends on the magnetic attraction between the units, while the binding force along the radial direction includes sliding friction which is related to the contact surface conditions. Conversely, for the side-to-side assembling, the binding force along the radial direction depends on the magnetic attraction between the units, while the binding force along the long axis includes sliding friction which is related to the contact surface conditions. In order to eliminate the error caused by the difference of contact surface conditions, we measure the force along the long axis under the end-to-end assembling and the radial force under the side-to-side assembling respectively for investigating the magnetic attraction between units under different magnetic fields.

5. Locomotion: Authors claim that the locomotion of micro robots on wet surfaces is difficult due to the surface tension, that's also the meaning of constructing the walking robot. For some biomedical applications that wet surfaces are everywhere. Does the actuation and assembly of micro units remain effective on wet surfaces?

Response: As we demonstrate in Fig. S7C and S7D (Fig. R2-1), the motion of unit monomer on the wet surface is also achievable. However, compared to bipedal walking, other morphologies and motion behaviors will line contact with wet surfaces inevitably, which may result in locomotion hindering and unintended structure separation since the greater resistance from surface tension. Therefore, to achieve the actuation and assembly of the units on wet surface, a stronger actuation magnetic field is required, and the motion error caused by the out-of-step should be considered.

Fig. R2-1. Performance comparison between different motion behaviors on wet surface. A, The motion of mCEBOT in wet surface is still challenging due to the viscosity and tension from surface, and the inevitable line contact in slipping or rolling. **B,** The resistance from surface tension can be greatly reduced benefited from the alternated point contact of biped structured mCEBOT.

6. Function: Despite the demonstration of adaptive multi-region releasing of CEBOT, the discussion of its potential application is missing. I believe this programmable and selective releasing by assembling the units with different decomposition properties could be an advantage of heterogeneous modular robot.

Response: Thanks for your comments. To avoid the overclaim, we have changed the application demonstration of in vitro bioengineering into engineering area based on the results obtained so far, including the taking hanging targets (main text page 13, Fig. 4) and the exploration of unstructured environments (main text page 14, Fig. 5). For the material enabled programmability, selective release and potential applications, we have discussed these in "Discussion section" (main text page 16).

7. On the decomposition of robotic units, are there any other methods instead of applying the magnetic stirring?

Response: Thanks for your comments. The decomposition triggering and rate of the unit are mainly determined by the environmental pH. During which, the decomposition process can be further accelerated by increasing the temperature or applying agitation. In our manuscript, we employ magnetic stirring to speed up the degradation process, as this is one of the easiest and most efficient methods for magnetic millirobots.

8. Besides the environmental pH and materials, the environmental temperature will also influence the decomposition process. The detailed environmental conditions should be provided in the investigation of decomposition.

Response: As reviewer said, the environmental temperature does have an effect on the decomposition process. Our experiment results indicate that, when keeps the pH constant, the higher the temperature, the faster the decomposition rate of the units. Since the biomedical applications in previous version have been replaced by engineering applications without unit decomposition, we will not discuss the decomposition in detail.

9. Authors are suggested to discuss the differences between the assembly of single CEBOT and multiple CEBOTs in a large number of robotic units.

Response: For the assembling of single mCEBOT, we continue to assemble the obtained assembly as a whole unit with other units. While, for the assembling of multiple mCEBOTs, we can re-select two new units to start the assembly of a new mCEBOT after the previous mCEBOT has been assembled. These related discussions have been provided in main text page 9.

Modifying:

Heterogeneous assembly enabled multiple configurations of morphologies

Besides that, the controllable and selective heterogeneous assembly can also enable the construction of single mCEBOT (Fig. 2I) or multiple mCEBOTs (Fig. 2J) in a large number of units, which provides more options for the construction of mCEBOT. When we continue to assemble the obtained assembly as a whole unit with other units, we can construct a single mCEBOT with a large number of units. While, when we re-select new units to start the assembly of a new mCEBOT after the previous mCEBOT has been assembled, we can construct multiple mCEBOTs.

10. What kind of Iron micro powder is used? Reference to biocompatibility is necessary especially when discussing its dissolution in biomedical applications.

Response: The iron micro powder used in experiment is purchased from Alfa Aesar (Spherical, APS6-10 micron, reduced, 99.5%). Since Fe is a natural metal (ferritin) in human body, we can metabolize iron particles into our elements [22]. Iron is safe for the individual within a moderate range and more biocompatible than other magnetic particles including Ni as well as Co [23]. By now, iron and its oxides have been widely used in biomedical applications, such as magnetic microrobot for drug delivery [24,25] and peripheral stent in tissue engineering [26].

11. Details on what bars, box or whiskers represent should be added in figure legends.

Response: Thanks, we have added the details on bars, box or whiskers in figure legends.

12. The grammar and style of writing have some errors throughout the manuscript and need to be checked and corrected. This will improve the work greatly and will make it more readable. For instance,

- a. "Benefit from" at the beginning of the sentence should be "Benefiting from";
- b. "different height" should be "different heights";
- c. "separates on-demand (at 123.5 s)" should be "separate on-demand (at 123.5 s)";
- d. "After that, we separate it into individual units by decreasing magnetic field strength to 0 mT, and accelerate the decomposition by applying magnetic stirring (50mT, 120 r/min)." a comma is missing;
- e. "the architecture is design properly" should be "the architecture is designed properly".

Response: We thank the reviewer for pointing out the language errors that are not fully corrected in the revised manuscript. We have carefully rewritten the abstract and fixed all of the language issues we found in the rest part of the manuscript.

Reviewer#3:

This paper some interesting points and design tricks, such as a hard magnetic to bond soft magnetic modules as a master-slave system. However, there are many similar works, and the authors should clearly define the real progress of their work.

Response: We thank the referee very much for his/her valuable comments. The main contribution of our work is the propose of heterogeneous assembling (modular units with soft magnetic material, frustum shape and slightly different dimensions) for constructing modular millirobot with simultaneously reconfigurable morphologies (independent monomer, bastinade shape, biped structure and hoe shape) and motion behaviors (slipping, rolling, walking and climbing). We have defined the real progress of our work by comparing with other similar works, and revised the manuscript with the most seriousness. Please find our detailed responses below.

1. A few magnetic modules can arrange with different shapes, which is not surprising. The demonstrations in Fig. 3 and 4 only show 3 modules, so the method does not appear to be scalable. The authors claim that their robotic system has advantages for navigating in unconstructed environments, and "the unstructured environment" in (Fig 4) is a well designed wet surface, a narrow space, and around an obstacle with fined tuned manual control and visual feedback.

Response: Thanks for your suggestion. As reviewer mentioned, there are some works can achieve the assembly of individual units into different constructs [1-14]. However, when extending the existing methods to the construction of modular millirobots, they will show more or less limits. The challenges include the selective actuation and control of independent unit [1-6], the controllability and reversibility of assembling process [7,8], multiplex connecting modes [1-6], rich expansion of motion behaviors and functions [9-14]. In this revised version, we have highlighted and demonstrated the main novelty of our work, i.e., heterogeneous assembly, simultaneous reconfiguration of morphologies and behaviors, and corresponding advantages in engineering tasks demonstration. Please check main text page 2-3 and page 14 for details.

For the scalability, we think it's not an issue. In fact, the scalability of our mCEBOT has been proved in Fig. 2G, 2H, 2I and 2J by adopting more than 3 units (Fig. R1-1). Our modular mCEBOT mainly focuses on the reconfiguration of morphologies and behaviors by adjusting the assembling modes and order of limited number of units, which is different from the indiscriminate virtual assembly and massive units in magnetic swarm. On the other hand, despite the diverse achievable morphologies of our mCEBOTs, they can be classified into four typical configurations, i.e., independent monomer, bastinade shape, biped structure and hoe mode, with the attributes of the smallest indivisible unit, largest aspect ratio, multi-point contact and hook anchor, respectively. Based on that, we adopted a minimum number of 3 units and the corresponding most basic typical structure for the demonstrations in Fig. 3 and Fig. 4.

To demonstrate the environmental adaptability of different motion modes more intuitively while keep its generic, we try to select common and representative terrains or environments for experiments. For example, when investigating the motion behaviors independently (Fig. 3), we adopt the narrow interior space of a common and existing plastic pipe to verify the effective slipping of mCEBOT in a small space, and use its larger

outer wall to verify the obstacle-crossing performance of mCEBOT rolling. However, it is not easy to find a suitable scene, including wet surface, narrow space, high barrier and hanging target at the same time, for multi-modal motion exhibition. In order to demonstrate the different motion modes and the transition process between them intuitively, we construct this specific scene in Fig. 4. Although the constructed scene is specific, the motion adaptation of mCEBOT is generic even if the obstacles or terrains varies within a reasonable range in the actual environment. For example, the barriers or targets with different heights and water films with different depths, can all be adapted by increasing or decreasing the assembling number of units. This is also verified by the unspecific and complexed scene we demonstrated in Fig. 5.

Fig. R1-1. The scalability of mCEBOT. **A**, Protean morphologies achieved by introducing units' number and combining two assembling methods. **B**, Multiple configurations of mCEBOT's morphologies by controllable assembly and separation. **C**, The construction of single mCEBOT in a large number of robotic units. **D**, The construction of multiple mCEBOTs at the same time in a large number of robotic units.

Modifying:

Introduction, main text page 2-3:

However, under the critical constraints of space and energy consumption, constructing modular robot at small-scale is still very difficult limited by the lack of conscious approaching, diverse docking, and selective detaching strategies between micro units (Table 1). Although swarm design can cluster massive micro/nano particles to perform certain collective behaviors [1-3], drawbacks in poor monomer control, non-selective unit approaching, and virtual modular connection make it only works under the global magnetic actuation. Moreover, despite the diverse achievable patterns of swarm, its motion behaviors are limited and mostly confined to the liquid environment. Applying additional frameworks, restriction or assistance [9-14] can also collect or assemble small units as desired, however these strategies impose more or less restrictions on the reversibility of assembly and the application of robot. Furthermore, most of these strategies only focus on the morphological reconfiguration and ignore the diversity of behaviors and functions of robot, which makes the small-scale modular robot far from reliable in practice.

Table 1. The comparison of existing assembly methods for small-scale modular robot

	Independent control units	Reversibility of assembly	Morphologies reconfiguration	Behaviors reconfiguration	Features	References
Self-assembly	-	Poor	Poor	Poor	Assembly process is uncontrollable	[5,6]
Irreversible assembly	-	-	-	Poor	Non-reversing	[7,8]
Virtual assembly	-	Good	Good	Good	Global actuation and limited to liquid environment	[1-3]
Assisted assembly	Good	Poor	Good	Poor	Less diversity in behaviors and functions	[9-14]
Heterogeneous assembled mCEBOT	Good	Good	Good	Good	Reconfigure morphologies and behaviors simultaneously	This work

1. “-” means cannot achieve; “Poor” means few can achieve; “Good” means most can achieve.

Locomotion adaptability in complexed environment, main text page 13:

Fig. 4. Combined multimodal locomotion modes of mCEBOT in a compound task. **A**, Schematic diagram of the complexed environment, where mCEBOT starts from the wet surface, and then needs to through a mezzanine and a barrier, finally has to take the hanging target. **B**, Experimental results of mCEBOT in the compound task, mainly including walking on wet surface, separating into separated state, slipping through narrow space, assembling into bastinade shape, rolling overcome obstacle, reassembling into hoe shape and taking hanging target. **C**, Time-lapse images of mCEBOT during hanging target taking. **D**, Height changing of mCEBOT and target during the hanging target taking task.

To further elaborate the environmental adaptability of mCEBOT, we set up an complexed environment containing wet surface (~300 m water film), narrow slit (mezzanine with

~0.7 mm gap, ~58% of long unit's height), obstacle area (barrier with ~4.0 mm height, ~4.7 times of short unit's height) and hanging target (~3.5 mm height, ~1.1 times of the maximum height of mCEBOT), as shown in Fig. 5A and Fig. 5B. To cross the wet surface, these individual units are reconfigured to biped structure firstly. Under the action of spatially varying magnetic field (100 mT), the assembled biped mCEBOT moves forward ~30 mm in 70 s and finally step out of the scope of water coverage in 80 s by walking motion. To cross the narrow slit, the biped structure is firstly separated into individual units by decreasing the magnetic strength to 0 mT at 84 s. Then, driven by an oscillating magnetic field (100 mT, 1 Hz), these independent robot monomers slip across the narrow mezzanine at 410 s by slipping motion. When meeting the high barriers, these individual units are reconfigured to bastinade shape by end-by-end assembling in 30 s (from 410 s to 440 s). As a result, under a rotating magnetic field (100 mT, 0.1 Hz), the assembled bastinade shaped mCEBOT overcomes the barrier at 452 s by rolling. In order to take the hanging target that is higher than robot itself height, the bastinade shaped mCEBOT is reassembled into hoe shape at 584 s. As the time-lapse images (Fig. 4C) and the height changing (Fig. 4D) shown, the target object is hanged on a vine-like branch with a ground clearance of ~3500 μm which is ~1.1 times of the maximum height of bastinade shaped mCEBOT. Despite mCEBOT cannot jump or fly, by taking a 1500 μm height vine-like branch as pivot, its reachable height can increase to ~4000 μm (~125% of initial height) under the climbing motion. Since the enhanced reachable height already meets demand, the hanging target is successfully taken in 50 s. These results suggest that the environmental adaptability of robot at small-scale could be greatly enhanced through morphologies and behaviors reconfigurations (movie S3), offering great opportunities for tackling the tasks in harsh environments.

mCEBOT for environment exploration and path marking, main text page 14:

Benefiting from the heterogeneous assembly endowed selective assembling and detaching of units, our mCEBOT can perform the tasks that conventional milli-scale robot cannot. For example, we can apply mCEBOT for unstructured environment exploration and path marking. On the one hand, mCEBOT can easily overcome complex terrain by the reconfiguration of morphologies and behaviors simultaneously. On the other hand, it can also realize the marking of valid paths by selective and stepwise unloading units, which can guidance the locomotion of other robots and improve their efficiency in passing through unstructured and complexed environments.

As shown in Fig. 5A, the environment explored as demo can be seen as a maze with complex terrain. During the mission, mCEBOT not only needs to overcome the complex terrain on the road (mainly exemplified by high barriers and narrow slits), but also needs to choose the correct route among the many forks to reach the destination. In order to keep the environmental adaptability of mCEBOT and mark valid paths in detail, the initial number of units of mCEBOT needs to be determined according to the complexity of the environment and tasks. On the one hand, too few initial units will limit the reconfiguration of morphologies and behaviors, and may not have enough marks for path marking. On the other hand, the number of units is not the more the better because the number of units far exceeding the demand will cause a burden on the control and actuation. After evaluation, the initial number of units in our demonstration is set as 5, and one of the units is left at the starting point as the first marker. When a high barrier is encountered, the remaining units will assemble into bastinade shape and quickly overcome it by rolling motion (Fig. 5B). While for a narrow slit, the units will remain separated state and traversing it by slipping motion (Fig. 5C). Benefiting from the selective assembly and the use of locally actuation magnetic field, mCEBOT can unload partial units as markers at desired sites.

For example, mCEBOT can separate a unit at the fork or before the obstacle. If the route is correct or the obstacle is successfully surmountable, then the detached unit will be preserved as a mark, otherwise the detached unit will be recycled when mCEBOT returns. Based on this strategy, our mCEBOT finally passed 4 obstacle areas (2 high barriers, 2 narrow slits) within 450 s and left 5 markers (3 for forks, 1 for starting point, one for destination) during the exploration process. To verify the effectiveness of environment exploration and path marking, we apply a new unit to conduct path following and marks recycling. As shown in Fig. 5D, the new unit can successfully reach the destination within 256 s under the guidance of the markers. In addition to the guiding role, the marks can also be recycled to assemble with new unit to enhance its environmental adaptability. Fig. 5E and 5F are the motion trajectories of each unit of mCEBOT in the process of path marking and path following. By comparison, it can be found that under the guidance of markers, the robot can omit the attempt of invalid paths when passing through the same unstructured environment, and reach the destination faster and more efficiently.

Fig. 5. Demonstration of mCEBOT for environment exploration and path marking. A, Schematic diagram of maze route with complex terrain. B, Barrier overcoming by assembling units into rolling mCEBOT, and corresponding path marking during exploration. C, Slit crossing by units slipping with separated state, and corresponding path marking during exploration. D, Path following of new unit, and corresponding markers recycling. E, The trajectory of mCEBOT units during environment exploration and path marking. F, The trajectory of mCEBOT units during path following and markers recycling.

2. I wonder what will happen if it is put it in a real intestine, and what level of adaptability the system will show.

Response: We fully agree that the in vivo animal test is a key step to prompt the robot to biomedical engineering. However, limited by existing in vivo imaging and actuation strategies, there are no reconfigurable magnetic microrobots for animal or human experiments [1-14]. This work focuses on the construction of modular magnetic robot at millimeter scale by heterogeneous assembling, and its potential advantages and applications endowed by the reconfigurable morphologies and motion behaviors. To avoid the overclaim, we have changed the application demonstration of in vitro bioengineering into engineering area based on the results obtained so far, including the taking hanging targets (main text page 13, Fig. 4) and the exploration of unstructured environments (main text page 14, Fig. 5). Compared with the current magnetic swarm robots, which can only move in the water environment or liquid surface [1-4], the multi-modal motion exhibited by our mCEBOT will be more suitable for the complex solid-liquid mixing of the esophagus, gastrointestinal tract, etc. environment in the future in vivo experiment.

3. Finally, is it overall better than the soft robot with multimodal locomotion (Metin Sitti's Nature paper)? I am afraid not, as that robot can jump.

Response: Thanks for your comments. Modular robot (CEBOT) and single unit robot are two different kinds of robot construction strategies. Despite they may show similar locomotion, they are fundamentally different. For instance, compared with the single unit robot, not only the fantastic robot mentioned by you, CEBOT has several unique properties due to its separable and reorganizable structure:

- 1) The CEBOT could achieve richer morphologies by the heterogeneous assembling and separating of modular units. That is different from the single unit robot which can only perform simple bending or stretching deformation restricted by the inherent physical structure.
- 2) The CEBOT could have greater adaptabilities with large-scale spanned. For example, it can easily cross the slit with the smallest unit size by separating, and overcome the obstacles more than ten times the size of the unit by assembling.
- 3) The modularity design of CEBOT could bring new possibilities to perform more complex tasks, such as modular replacement for self-healing, partial separation and recycle for navigation, stepwise degradation and release for multi-point cargo delivery, etc.

In this revised version, we have added more data to demonstrate the unique advantages of modular mCEBOT, such as in complexed environment task and environment exploration. Please check main text page 13-14.

References:

1. Xie, H., Sun, M., Fan, X., Lin, Z., Chen, W., Wang, L., ... & He, Q. (2019). Reconfigurable magnetic microrobot swarm: Multimode transformation, locomotion, and manipulation. *Science Robotics*, 4(28).
2. Yu, J., Jin, D., Chan, K.F., Wang, Q., Yuan, K. and Zhang, L., 2019. Active generation and magnetic actuation of microrobotic swarms in bio-fluids. *Nature communications*, 10(1), pp.1-12.
3. Wang, Q., Chan, K.F., Schweizer, K., Du, X., Jin, D., Yu, S.C.H., Nelson, B.J. and Zhang, L., 2021. Ultrasound Doppler-guided real-time navigation of a magnetic microswarm for active endovascular delivery. *Science Advances*, 7(9), p.eabe5914.
4. Gardi, G., Ceron, S., Wang, W., Petersen, K. and Sitti, M., 2022. Microrobot collectives with reconfigurable morphologies, behaviors, and functions. *Nature Communications*, 13(1), pp.1-14.
5. Lu, H., Liu, Y., Yang, Y., Yang, X., Tan, R. and Shen, Y., 2018. Self-assembly magnetic chain unit for bulk biomaterial actuation. *IEEE Robotics and Automation Letters*, 4(2), pp.262-268.
6. Snezhko, A. and Aranson, I.S., 2011. Magnetic manipulation of self-assembled colloidal asters. *Nature materials*, 10(9), pp.698-703.
7. Cheng, Y., Chan, K.H., Wang, X.Q., Ding, T., Li, T., Zhang, C., Lu, W., Zhou, Y. and Ho, G.W., 2021. A fast autonomous healing magnetic elastomer for instantly recoverable, modularly programmable, and thermorecyclable soft robots. *Advanced Functional Materials*, 31(32), p.2101825.
8. Kuang, X., Wu, S., Ze, Q., Yue, L., Jin, Y., Montgomery, S.M., Yang, F., Qi, H.J. and Zhao, R., 2021. Magnetic dynamic polymers for modular assembling and reconfigurable morphing architectures. *Advanced Materials*, 33(30), p.2102113.
9. Alapan, Y., Yigit, B., Beker, O., Demirörs, A. F., & Sitti, M. (2019). Shape-encoded dynamic assembly of mobile micromachines. *Nature materials*, 18(11), 1244-1251.
10. Gu, H., Boehler, Q., Ahmed, D. and Nelson, B.J., 2019. Magnetic quadrupole assemblies with arbitrary shapes and magnetizations. *Science Robotics*, 4(35), p.eaax8977.
11. Diller, E., Pawashe, C., Floyd, S. and Sitti, M., 2011. Assembly and disassembly of magnetic mobile micro-robots towards deterministic 2-D reconfigurable micro-systems. *The International Journal of Robotics Research*, 30(14), pp.1667-1680.
12. Boyvat, M. and Sitti, M., 2021. Remote Modular Electronics for Wireless Magnetic Devices. *Advanced Science*, 8(17), p.2101198.
13. Zhang, J., Ren, Z., Hu, W., Soon, R.H., Yasa, I.C., Liu, Z. and Sitti, M., 2021. Voxellated three-dimensional miniature magnetic soft machines via multimaterial heterogeneous assembly. *Science robotics*, 6(53), p.eabf0112.
14. Diller, E., Zhang, N. and Sitti, M., 2013. Modular micro-robotic assembly through magnetic actuation and thermal bonding. *Journal of Micro-Bio Robotics*, 8(3), pp.121-131.
15. Timonen, J.V., Johans, C., Kontturi, K., Walther, A., Ikkala, O. and Ras, R.H., 2010. A facile template-free approach to magnetodriven, multifunctional artificial cilia. *ACS applied materials & interfaces*, 2(8), pp.2226-2230.
16. Demirörs, A.F., Aykut, S., Ganzeboom, S., Meier, Y.A., Hardeman, R., de Graaf, J., Mathijssen, A.J., Poloni, E., Carpenter, J.A., Ünlü, C. and Zenhäusern, D., 2021. Amphibious transport of fluids and solids by soft magnetic carpets. *Advanced Science*, 8(21), p.2102510.
17. Cao, M., Ju, J., Li, K., Dou, S., Liu, K. and Jiang, L., 2014. Facile and Large - Scale Fabrication of a Cactus - Inspired Continuous Fog Collector. *Advanced Functional Materials*, 24(21), pp.3235-3240.
18. Zhang, D., Wang, W., Peng, F., Kou, J., Ni, Y., Lu, C. and Xu, Z., 2014. A bio-inspired inner-motile photocatalyst film: a magnetically actuated artificial cilia photocatalyst. *Nanoscale*, 6(10), pp.5516-5525.
19. Azarmi, S., Farid, J., Nokhodchi, A., Bahari-Saravi, S.M. and Valizadeh, H., 2002. Thermal treating as a tool for sustained release of indomethacin from Eudragit RS and RL matrices. *International journal of pharmaceuticals*, 246(1-2), pp.171-177.
20. Aguilar, L.E., Unnithan, A.R., Amarjargal, A., Tiwari, A.P., Hong, S.T., Park, C.H. and Kim, C.S., 2015. Electrospun polyurethane/Eudragit® L100-55 composite mats for the pH dependent release of paclitaxel on duodenal stent cover application. *International journal of pharmaceuticals*, 478(1), pp.1-8.
21. Thakral, S., Thakral, N.K. and Majumdar, D.K., 2013. Eudragit®: a technology evaluation. *Expert*

- opinion on drug delivery, 10(1), pp.131-149.
22. H. Markides, M. Rotherham and A.J. El Haj, 2012. Biocompatibility and toxicity of magnetic nanoparticles in regenerative medicine. *Journal of Nanomaterials*, 2012.
 23. B. Liu and Y.F. Zheng, 2011. Effects of alloying elements (Mn, Co, Al, W, Sn, B, C and S) on biodegradability and in vitro biocompatibility of pure iron. *Acta biomaterialia*, 7(3), pp.1407-1420.
 24. S.Y. Chin, Y.C. Poh, A.C. Kohler, J.T. Compton, L.L. Hsu, K.M. Lau, S. Kim, B.W. Lee, F.Y. Lee and S.K. Sia, 2017. Additive manufacturing of hydrogel-based materials for next-generation implantable medical devices. *Science robotics*, 2(2).
 25. H. Lu, M. Zhang, Y. Yang, Q. Huang, T. Fukuda, Z. Wang and Y. Shen, 2018. A bioinspired multilegged soft millirobot that functions in both dry and wet conditions. *Nature communications*, 9(1), pp.1-7.
 26. Peuster, M., Hesse, C., Schloo, T., Fink, C., Beerbaum, P. and von Schnakenburg, C., 2006. Longterm biocompatibility of a corrodible peripheral iron stent in the porcine descending aorta. *Biomaterials*, 27(28), pp.4955-4962.
 27. Kim, J., Chung, S.E., Choi, S.E., Lee, H., Kim, J. and Kwon, S., 2011. Programming magnetic anisotropy in polymeric microactuators. *Nature materials*, 10(10), pp.747-752.

REVIEWERS' COMMENTS

Reviewer #1 (Remarks to the Author):

The manuscript should be accepted for publication. The authors made a significant improvement in the revised manuscript by clarifying their contributions in detail and showing the unique advantage of modular mCEBOT in performing tasks in complex environment. The authors fairly addressed all previous questions with additional new experiments.

Some minor comments related to the manuscript:

- 1) On page 13 line 4; Fig. 5A and Fig. 5B should be Fig. 4A and Fig. 4B.
- 2) On page 14 line 5; "can guidance..." should be written as "can guide...".
- 3) On page 14 line 10; "but also needs to choose" should be written as "but also choose".
- 4) On page 14 line 20; similarly should be "will remain in..."

Reviewer #2 (Remarks to the Author):

In this revised version, the author removed the bio-application contexts and added more experiments and discussions to focus on the concept and construction of the milli-modular robot itself. Compared with the previous version, the novelty and contribution of the manuscript is much clearer. Particularly, the new added videos are cool and clearly exhibit the unique features and abilities of the modular robot compared with the existing soft robots and swarm robots, etc. In this regard, all my concerns to the previous version have been well addressed. I suggest it for publication in NC and wish it can bring new inspirations to the researchers and publics.

Some minor commons to the author:

1. The author said that "the frustum shape can endow unequal arrangements gap for reducing internal magnetic repulsion", but the description here is not very clear. Please provide more details.
2. Please discuss and prospect more specific applications of the selective assembling and detaching feature of modular units.
3. The abbreviations "mCEBOT" should be unified in main text and figures.

Reviewer#1:

The manuscript should be accepted for publication. The authors made a significant improvement in the revised manuscript by clarifying their contributions in detail and showing the unique advantage of modular mCEBOT in performing tasks in complex environment. The authors fairly addressed all previous questions with additional new experiments.

Response: We thank the referee very much for his/her recognition of our revised version. We also thank the referee for his/her recommendation for publish our work in Nature Communications.

1. Some minor comments related to the manuscript: On page 13 line 4; Fig. 5A and Fig. 5B should be Fig. 4A and Fig. 4B.

Response: Thanks for your careful reading. According to the suggestion of editor, we have reorganized the order of figures and double check the figure citing in the revised version.

2. On page 14 line 5; "can guidance..." should be written as "can guide...".

Response: Thanks. "can guidance..." on page 14 line 5 has been changed into "can guide...".

3. On page 14 line 10; "but also needs to choose" should be written as "but also choose".

Response: Thanks for your suggestion. We have revised "but also needs to choose" on page 14 line 10 as "but also choose".

4. On page 14 line 20; similarly should be "will remain in..."

Response: Thanks. The missing preposition "in" has been added in the revised version.

Reviewer#2:

In this revised version, the author removed the bio-application contexts and added more experiments and discussions to focus on the concept and construction of the milli-modular robot itself. Compared with the previous version, the novelty and contribution of the manuscript is much clearer. Particularly, the new added videos are cool and clearly exhibit the unique features and abilities of the modular robot compared with the existing soft robots and swarm robots, etc. In this regard, all my concerns to the previous version have been well addressed. I suggest it for publication in NC and wish it can bring new inspirations to the researchers and publics.

Response: We thank the referee very much for the review and high recognition of our work. We also thank the referee for his/her recommendation for publish our work in Nature Communications.

1. Some minor commons to the author: The author said that “the frustum shape can endow unequal arrangements gap for reducing internal magnetic repulsion”, but the description here is not very clear. Please provide more details.

Response: Thanks for your comments. In the revised version, we have added more detailed descriptions to explain the unique design of frustum shape and its function in reducing internal magnetic repulsion.

Modifying:

RESULT: Units design and characterization

Particularly, we design the units to frustum shape with a large aspect ratio and two differentiated section radiuses. Here the large aspect ratio of unit could guide the magnetic poles distributing to two ends to reduce the probability of unpredictable and uncontrollable assembly. And the two differentiated section radiuses can endow unequal arrangements gap during multiple side-by-side assemblies for reducing internal magnetic repulsion, which enables the easier and more stable connection than the uniform cylinder with the same size.

2. Please discuss and prospect more specific applications of the selective assembling and detaching feature of modular units.

Response: Thanks for your suggestion. The specific potential applications of the selective assembling and detaching has been added in discussion section.

Modifying:

DISCUSSION

Thirdly, the selective assembling and detaching of units make mCEBOT can on demand leave some units for complex tasks, such as the active retention of mCEBOT's units during the environment exploration for path marking, the localized separation of mCEBOT's units (with cargo inside) in different target areas for multi-regional delivery, the stepwise degradation of mCEBOT's units (with different drugs inside) in GI tract for combination therapy.

3. The abbreviations "mCEBOT" should be unified in main text and figures.

Response: We thank the referee very much for his/her suggestions. The "cebot" in figure and main text has been changed into "mCEBOT".